# Quantum annealing for neural network optimization problems: A new approach via tensor network simulations

Guglielmo Lami[1*], Pietro Torta[1], Giuseppe E. Santoro[1,2,3] and Mario Collura[1,4]

**1** SISSA, Via Bonomea 265, I-34136 Trieste, Italy
**2** International Centre for Theoretical Physics (ICTP), P.O. Box 586, I-34014 Trieste, Italy
**3** CNR-IOM Democritos National Simulation Center, Via Bonomea 265, I-34136 Trieste, Italy
**4** INFN Sezione di Trieste, via Bonomea 265, I-34136 Trieste, Italy

⋆ glami@sissa.it

## Abstract

Here, we focus on the problem of minimizing complex classical cost functions associated with prototypical discrete neural networks, specifically the paradigmatic Hopfield model and binary perceptron. We show that the adiabatic time evolution of QA can be efficiently represented as a suitable Tensor Network. This representation allows for simple classical simulations, well-beyond small sizes amenable to exact diagonalization techniques. We show that the optimized state, expressed as a Matrix Product State (MPS), can be recast into a Quantum Circuit, whose depth scales only linearly with the system size and quadratically with the MPS bond dimension. This may represent a valuable starting point allowing for further circuit optimization on near-term quantum devices.



# 1   Introduction

A large number of relevant computational and physical problems can be equivalently re-cast to a non-convex optimization task of a suitable Ising-like cost function [1] in the form $H(\boldsymbol{\sigma}) = H(\sigma_1, \ldots, \sigma_N)$, depending on $N$ binary variables $\sigma_i = \pm 1$. Since an exact solution implies the daunting task of a discrete search in a space with exponentially-growing dimension in the system size $N$, the optimization is usually carried out approximately, by means of heuristic minimization algorithms. Among these, a growing body of interest is devoted to quantum optimization, which aims at exploiting quantum effects, namely quantum superposition and entanglement, to obtain a wave function with a large overlap with classical solutions.

Alongside with well-established schemes such as Quantum Annealing (QA) [2–6] and Adiabatic Quantum Computation (AQC) [7,8], implemented in analogue dedicated hardware [9], recent approaches encompass the design of parameterized quantum circuits, implemented on a digital quantum device, which are run in loop with a classical computer in Variational Quantum Algorithms (VQA) [10].

A conceptual preliminary step in quantum optimization is to map classical spins on quantum spin-1/2 Pauli operators $\hat{\sigma}_j^z$, hence regarding the initial cost function as a quantum Hamiltonian that is *diagonal*, by construction, in the standard computational basis of quantum computation [11]:

$$H(\sigma_1, \ldots, \sigma_N) \to \hat{H}_z(\hat{\sigma}_1^z, \ldots, \hat{\sigma}_N^z). \tag{1}$$

Next, in QA/AQC a non-commuting driving term, often a transverse field, is introduced, and the quantum Hamiltonian is taken to be

$$\hat{H}(s) = s(t)\hat{H}_z + \big(1 - s(t)\big)\hat{H}_x, \qquad \hat{H}_x = -\sum_{i=1}^{N} \hat{\sigma}_i^x, \tag{2}$$

with an interpolation parameter $s(t) \in [0,1]$ such that $s(0) = 0$ and $s(\tau) = 1$, $\tau$ being the total annealing time. If $\tau$ is large enough — compared to the inverse square of the minimal spectral gap along the driving — one can rely on adiabatic theorems [8] to prove that the system will be driven into the ground state of the target Hamiltonian $\hat{H}_z$.

In its digitized version [12–14] (dQA), the QA Schrödinger dynamics is implemented step-wise, in P discrete time steps of length $\delta t = \tau/P$, after Trotter splitting the two non-commuting terms. This leads to a quantum state of the form

$$|\psi_P\rangle = e^{-i\beta_P \hat{H}_x} e^{-i\gamma_P \hat{H}_z} \cdots e^{-i\beta_p \hat{H}_x} e^{-i\gamma_p \hat{H}_z} \cdots e^{-i\beta_1 \hat{H}_x} e^{-i\gamma_1 \hat{H}_z} |\psi_0\rangle, \tag{3}$$

where $|\psi_0\rangle = |\rightarrow\rangle^{\otimes N}$ is the ground state of $\hat{H}_x$, with $|\rightarrow\rangle = \frac{1}{\sqrt{2}}(|\uparrow\rangle + |\downarrow\rangle)$). Assuming a linear annealing schedule $s(t) = t/\tau$, and to lowest order in the Trotter split-up, the parameters $\boldsymbol{\beta} = (\beta_1 \cdots \beta_{\mathrm{P}})$ and $\boldsymbol{\gamma} = (\gamma_1 \cdots \gamma_{\mathrm{P}})$ are given by $\beta_p = (1 - p/\mathrm{P})\delta t$ and $\gamma_p = (p/\mathrm{P})\delta t$, with $p = 1 \cdots \mathrm{P}$. In turn, by regarding $\boldsymbol{\beta}$ and $\boldsymbol{\gamma}$ as 2P variational parameters, this is the starting point of a VQA [10] approach, notably the Quantum Approximate Optimization Algorithm (QAOA) [15].

Despite promising results in problem-specific settings [8], the actual effectiveness and scalability of quantum optimization schemes for classical optimization is still debated. In fact, the quest for quantum speed-ups [16, 17] and the real effectiveness of quantum optimization algorithms should ultimately be tested on real scalable quantum devices, beyond the reach of classical simulations by means of Exact Diagonalization (ED) techniques. To implement this program, however, one encounters two main hurdles.

The first, concerns available experimental platforms for quantum devices: despite major progresses and steady development [18], the number of available physical qubits and their connectivity are quite limited. Moreover, experimentally available qubits are very sensible to noise, thus limiting realistic applications to shallow circuits requiring short coherence times. These technical issues severely limit, in practice, the feasibility of quantum simulations beyond the classical limits.

Secondly, the actual implementation of QA [9] on analogue devices, as well as that of digitized Quantum Annealing (dQA) [12] or VQAs [10] on a digital circuit-based quantum computer, usually requires an actual implementation of the unitary time evolution generated by the quantum Hamiltonian in Eq. (1). This often constitutes a formidable technical challenge: while few problems such as Max-Cut on regular graphs [19] only involve two-body interactions, directly implementable in an analogue/digital device, general optimization tasks usually yield a Hamiltonian $\hat{H}_z$ with non-local multi-spin interactions, hence difficult to implement. These experimental limitations and theoretical challenges call for efficient classical simulations of quantum optimization protocols, beyond the usual small-scale limits imposed by ED techniques, thus overcoming the so-called "curse of dimensionality" of an exponentially-large Hilbert space. Moreover, it would be particularly interesting to assess the performance of quantum optimization algorithms for those models bearing hard instances of Hamiltonians $\hat{H}_z$, which are not immediately encoded into a small set of single and two-qubits gates available on present-day quantum devices. A prominent family of classical simulation techniques allowing for large-scale simulations of quantum systems is represented by Tensor Networks. Major results in this framework include winning strategies for one-dimensional quantum many-body systems [20–25] and, more recently, significant contributions in Machine Learning [26–28] and hybrid quantum-classical algorithms [29]. The goal of Tensor Networks is to provide an efficient representation of quantum many-body wave functions in the form of a generic network of tensors, connected by means of auxiliary indices [21, 30]. These indices are characterized by a fixed bond dimension, $\chi$, which controls the information content of the network, characterizing its ability to encode entangled states. Matrix Product States (MPS) [20] are the simplest class of Tensor Networks: an MPS wave function is obtained by the multiplication of site-dependent $(\chi \times \chi)$ matrices $A^{[i]}(\sigma_i)$, each depending only on the local spin variable $\sigma_i \in \{+1, -1\}$ (see Fig.1). MPS can be manipulated efficiently in classical numerical simulations by means of well-established algorithms, as the Density Matrix Renormalization Group (DMRG) [20].

In this paper, we present a novel framework to efficiently simulate quantum optimization algorithms for a large class of hard classical optimization problems. We focus on a standard dQA approach by repeatedly applying the unitary operators $e^{-i\gamma_p \hat{H}_z}$, $e^{-i\beta_p \hat{H}_x}$ and iteratively projecting back, at each step, the resulting state on the MPS manifold $\mathcal{M}_\chi$ with a fixed bond dimension $\chi$. A first main result of our work is a theoretical construction that yields an *effi-*

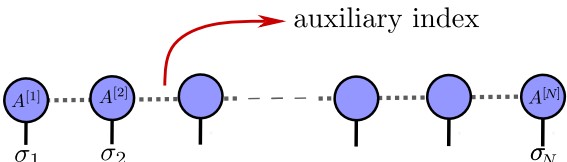

Figure 1: A graphical representation of a generic Matrix Product State (MPS). Dotted lines represent auxiliary (unphysical) indices, running from 1 to a fixed integer $\chi \geq 1$ (bond dimension). Solid lines represent the physical variables $\sigma_i \in \{+1, -1\}$. Notice that the first and the last tensors $A^{[1]}$, $A^{[N]}$ have only one auxiliary index, being respectively a row and a column vector.

*cient* MPS-based representation of dQA for a family of classical cost-function Hamiltonians $\hat{H}_z$, inspired by paradigmatic discrete neural networks and encompassing models with non-local multi-spin interactions, up to $N$-body terms. This results in an efficient algorithm, with computational cost scaling *polynomially* with the system size $N$, allowing for classical simulations well-beyond typical sizes analyzed by means of ED techniques.

Our numerical results are two-fold. First, in the regime of small time step $\delta t \ll 1$ — where dQA closely approximates the continuous QA dynamics — our approach is reliable, since it can systematically reproduce ED simulations of dQA, with a high degree of accuracy. Secondly, in the regime of large $\delta t \sim \mathcal{O}(1)$ — characterized by large Trotter errors that spoil the dQA accuracy — we observe that our algorithm can significantly outperform ED simulations of dQA, surprisingly providing far better-quality solutions for the optimization problem. We provide the following interpretation of this unexpected effectiveness (see sketch in Fig. 2 and Sec. 3.3 for a comprehensive discussion). The initial state $|\psi_0\rangle$ is a (trivial) MPS of bond dimension $\chi = 1$. Moreover, the final annealed state $|\psi_\tau\rangle$ — resulting from the exact QA time evolution (with $\tau \gg 1$) and thus expected to yield a large overlap with classical solutions — is often a low-entanglement state, efficiently represented by a MPS of low bond dimension $\chi$. This fact is certainly true for low-entangled many-body ground state preparation; nevertheless, it may also be verified in the context of classical optimization problems, whenever the number of classical solutions (spanning the ground-state eigenspace) is small enough, or when the exact QA converges to a cluster of solutions. These conditions are met for the models we examine, as detailed in Sec. 3.3 and Appendix E. Hence, in this case, both the initial and the final states of QA belong to the manifold $\mathcal{M}_\chi$, although the intermediate states may generally lay out of the manifold. As stated above, in the regime of small $\delta t$, results based on ED and on MPS simulations essentially coincide; on the contrary, in the regime of large $\delta t$, dQA faces the blowing up of the Trotter errors, leading to an ED dynamics governed by an effective Hamiltonian that substantially differs from the original one. The resulting final state may thus deviate from the classical solutions of the target Hamiltonian. In addition, it could encode unwanted entanglement due to the spurious terms generated by the Trotter splitting. Our MPS approach relies instead on multiple projections of the time-evolved state on $\mathcal{M}_\chi$ for each time step $\delta t$: this may explain, for some class of optimization problems, the enhanced effectiveness of our MPS-based algorithm, since its final state may be closer to the optimal annealed state $|\psi_\tau\rangle$. These findings provide a novel promising application of tensor network techniques, which might be adapted to implement efficient classical simulations for other quantum optimization algorithms.

Finally, we show that a gate-decomposition of the final annealed MPS, yields efficient quantum circuits that effectively solve hard classical optimization problems, with a number of basis gates that grows only linearly in the system size $N$ and quadratically with the MPS bond dimension $\chi$. This result not only yields a numerical proof of principle on the effectiveness of

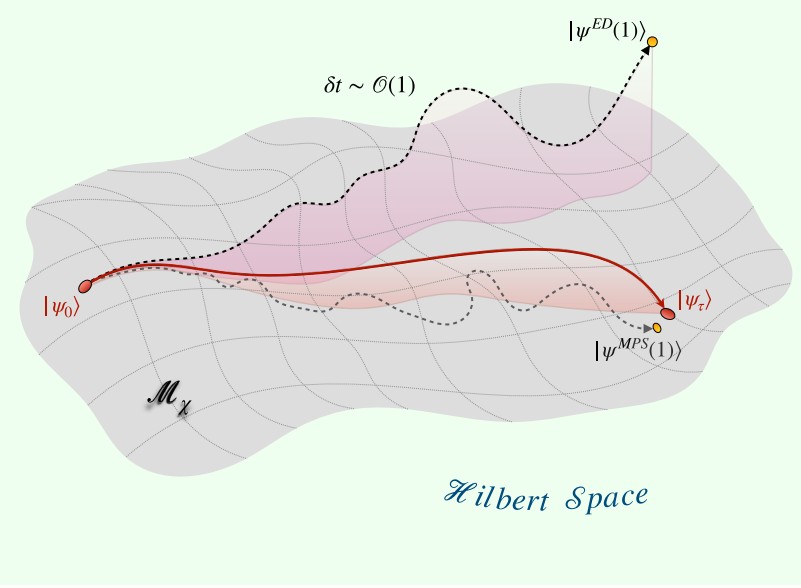

Figure 2: The MPS manifold $\mathcal{M}_\chi$, embedded in the exponentially larger Hilbert space. Different QA protocols are represented as trajectories (lines) in this space. The exact adiabatic dynamics (full red line) for a finite but large annealing time $\tau \gg 1$ is not generally constrained into the MPS manifold; nevertheless, it will lead from $|\psi_0\rangle$ (MPS with $\chi = 1$) to a final state $|\psi_\tau\rangle$, which also belongs to the MPS manifold. As soon as the adiabatic-theorem conditions are met, $|\psi_\tau\rangle$ is expected to yield a big overlap with the ground-state eigenspace (spanned by all classical solutions). However, when the true dynamical protocol is dQA with a finite time step $\delta t \sim \mathcal{O}(1)$, the actual Trotterized dynamics (black dashed line) may largely deviate from the true adiabatic one, due to an unwanted production of extra entanglement (see Figs. 12, 13 in Sec. 3.3 for a discussion of these aspects). Hence, the final state $|\psi^{ED}(1)\rangle$ may lay outside the MPS manifold, and it definitely differs from the final QA state $|\psi_\tau\rangle$, yielding very small overlap with classical solutions. Remarkably, when the Trotterized dynamics is performed within our novel framework (gray dotted line) — i.e. constrained to $\mathcal{M}_\chi$, by alternating the Trotter steps with projections into the manifold — the MPS evolved state remains closer to the exact dynamics, thus finally leading to a final state $|\psi^{MPS}(1)\rangle$ which is very close to the target state.

quantum circuits in these regimes, but may also serve as a guideline to develop new classes of parameterized quantum circuits.

The rest of the manuscript is organized as follows. In Section 2 we introduce the proto-typical neural network models tackled in this work, which fall into a large class of problems that can be studied with our techniques. We also provide a detailed explanation of our novel theoretical framework, enabling efficient simulations of dQA with MPS, for any model falling into this class. Section 3 is devoted to numerical results, first for a benchmark class of p-spin integrable models, and then for the prototypical neural network models. In the same Section, we provide numerical evidence supporting the theoretical interpretation detailed in Fig. 2, explaining the unexpected effectiveness of our methods for some class of problems. Finally, we conclude and discuss future perspectives stemming from our work.

## 2 Models and methods

### 2.1 Models

The classical optimization problems we analyze can be formulated as a ground-state search for a classical Hamiltonian $H(\boldsymbol{\sigma})$, which can be mapped in a quantum setting as outlined in Eq. (1). In our paper, we shall focus on a broad class of classical Hamiltonians (or cost functions) that can be cast in the following form:

$$H(\boldsymbol{\sigma}) = \sum_{\mu=1}^{N_\xi} h(\boldsymbol{\xi}^\mu \cdot \boldsymbol{\sigma}), \tag{4}$$

where the $\xi_i^\mu \in \{-1, +1\}$ ($\mu = 1, 2 \ldots N_\xi$, $i = 1, 2 \ldots N$) are (possibly random) spin configurations, usually called patterns, and $h$ is any sufficiently regular function. As anticipated, our methods, detailed in the following section, are quite general: they apply to any Hamiltonian that can be rewritten in this generic form, which is reminiscent of the cost function of simple discrete neural networks. Among many possible choices, we may focus on specific models proving particularly challenging to implement on a quantum device, for two different (possibly concurring) reasons: a required long/infinite range two-qubit gates connectivity and the presence of multi-spin interactions. In particular, let us notice that any non-quadratic function $h$ leads to spin-spin interaction terms beyond usual 2-bodies interactions.

As a preliminary benchmark for our strategy, we validate the results of our MPS-based technique against ED results for simple integrable p-spin models [31–33]:

$$H^{\mathrm{p-spin}}(\boldsymbol{\sigma}) = -N \left( \frac{1}{N} \sum_i \sigma_i \right)^{\mathrm{p}}, \tag{5}$$

which, for p = 2, is also known as the Lipkin-Meshkov-Glick (LMG) model (or infinite-range Ising model). Let us notice that the p-spin Hamiltonian can be rewritten as in Eq. 4 with a single pattern $\boldsymbol{\xi}^0 = (+1, +1, \ldots, +1)$ and $h(x) = -N^{1-\mathrm{p}} x^{\mathrm{p}}$. These benchmark models have trivial ground states: for even p, the classical ground-states are the two ferromagnetic states $\boldsymbol{\sigma} = \pm \boldsymbol{\xi}^0$ (with energy $E_{gs} = -N$) whereas for odd p only $\boldsymbol{\sigma} = +\boldsymbol{\xi}^0$ is a ground state (with energy $E_{gs} = -N$). Thanks to the integrability of p-spin models, we are able to verify the agreement of MPS and ED results up to large system sizes.

We then focus on two prototypical optimization problems coming from the realm of machine learning and Artificial Neural Networks (ANNs) [34, 35]. First, we consider the Hopfield model [34–41], a simple recurrent neural network studied in unsupervised learning

$$H^{\mathrm{Hopfield}}(\boldsymbol{\sigma}) = -\sum_{i,j} J_{ij} \sigma_i \sigma_j = -\sum_{\mu=1}^{N_\xi} \left( \frac{\boldsymbol{\xi}^\mu \cdot \boldsymbol{\sigma}}{\sqrt{N}} \right)^2, \qquad J_{ij} = \frac{1}{N} \sum_{\mu=1}^{N_\xi} \xi_i^\mu \xi_j^\mu. \tag{6}$$

In this context, the patterns $\{\boldsymbol{\xi}^\mu\}_{\mu=1}^{N_\xi}$ are i.i.d. random variables and the goal is to memorize them in the classical ground states. The Hopfield Hamiltonian entails infinite-range two-body interactions between any spin pair, and it can be seen as a non-integrable generalization of the p = 2-spin model. Concerning the relation between the number of patterns $N_\xi$ and the number of variables $N$, it is customary to set $N_\xi = \alpha N$ with $\alpha = \mathcal{O}(1)$. Different regimes/phases in the thermodynamic limit ($N \to \infty$) are distinguished by different values of $\alpha$. In particular, for the Hopfield model at zero temperature, $\alpha_c \simeq 0.138$ represents a critical value separating a retrieval phase in which the model works as a memory device ($\alpha < \alpha_c$), from a non-retrieval/spin-glass phase ($\alpha > \alpha_c$) [40].

Secondly, we examine the binary perceptron, the prototypical example of a single-layer binary classifier, which is a fundamental building block of ANNs routinely used in supervised learning [34, 35, 42]. In this case, the spin variables $\boldsymbol{\sigma}$ are identified with binary synaptic weights, classifying correctly a given pattern $\boldsymbol{\xi}^\mu$ into a prescribed binary label $\tau^\mu = \pm 1$ if $\text{sgn}(\boldsymbol{\sigma} \cdot \boldsymbol{\xi}^\mu) = \tau^\mu$. During the training phase, a given labeled data-set $\{\boldsymbol{\xi}^\mu, \tau^\mu\}_{\mu=1}^{N_\xi}$ is provided, and the objective consists in finding weight configurations $\boldsymbol{\sigma}$ that classify correctly the whole training set. This is naturally formulated as a minimization problem of a suitable cost function, which assigns a positive energy cost for every pattern incorrectly classified, with the exact solutions to the classification problem being characterized as zero-energy configurations. A common choice for the classical cost function is given by

$$H^{\text{perceptron}}(\boldsymbol{\sigma}) = \sum_{\mu=1}^{N_\xi} \theta\big(-\boldsymbol{\xi}^\mu \cdot \boldsymbol{\sigma}\big)\Big(\frac{-\boldsymbol{\xi}^\mu \cdot \boldsymbol{\sigma}}{\sqrt{N}}\Big), \tag{7}$$

where $\theta(x)$ is the Heaviside step function. In the previous expression, the labels $\tau^\mu$ are all set to 1, as it can be done without loss of generality for learning random patterns, i.e. if patterns and labels are both drawn from an unbiased Bernoulli distribution. Let us observe that the Hamiltonian in Eq. 7 is in the general form given by Eq. 4. Despite encouraging numerical and analytical evidence on the effectiveness of quantum optimization for this model [43, 44] and other closely related models [45], the perceptron Hamiltonian implies all possible interactions among spins, up to $N$-body terms, hence it is not efficiently implementable on a quantum device. Also for the perceptron model, it is customary to study the $N_\xi = \alpha N$ regime, with $\alpha = \mathcal{O}(1)$, since the critical capacity in the thermodynamic limit is $\alpha_c \simeq 0.83$ [46], separating a SAT region for $\alpha < \alpha_c$, admitting zero-energy solutions, from an UNSAT region $\alpha > \alpha_c$.

## 2.2 Digitized Quantum Annealing (dQA)

As briefly sketched in the Introduction, a standard procedure to implement Quantum Annealing on digital quantum simulators, or to simulate its dynamics classically, relies on a discretization of the continuous QA time evolution in $P \gg 1$ time steps of length $\delta t = \tau/P$, followed by a Trotter split-up of the two non-commuting terms.

Albeit this scheme can be easily generalized to higher orders, here we stick to the lowest-order contributions

$$e^{-i\hat{H}(s_p)\delta t} \simeq e^{-i(1-s_p)\hat{H}_x \delta t}\, e^{-i s_p \hat{H}_z \delta t} + \mathcal{O}\big(\delta t\big)^2, \tag{8}$$

where $p = 1, 2, \ldots, P$ and $s_p = t_p/\tau = p/P$. Introducing the shorthands $\beta_p = (1-s_p)\delta t$, and $\gamma_p = s_p \delta t$, we can rewrite the previous expression more concisely as

$$e^{-i\hat{H}(s_p)\delta t} \simeq \hat{U}_x(\beta_p)\,\hat{U}_z(\gamma_p) + \mathcal{O}\big(\delta t\big)^2, \quad \text{with} \quad \hat{U}_x(\beta_p) = e^{-i\beta_p \hat{H}_x}, \quad \hat{U}_z(\gamma_p) = e^{-i\gamma_p \hat{H}_z}, \tag{9}$$

thus recovering Eq. (3).

The dQA framework can reproduce accurately the real Quantum Annealing dynamics for any value of the total annealing time $\tau$, which is exactly recovered by simultaneously scaling $P \rightarrow \infty$ and $\delta t \rightarrow 0$, setting their product equal to $\tau$. In practice, for a fixed value of P, the optimal value of the Trotter step $\delta t$ depends on a trade-off between the Trotter errors and the annealing time $\tau$ [13]: for small $\delta t$ (small $\tau$) the time evolution is not adiabatic, whereas for large $\delta t$ (large $\tau$) the Trotter split-up is expected to become a rough approximation and to introduce spurious quantum correlations. As discussed in Sec. 3.3, the discrete-time evolution obtained without any Trotter split-up is often unexpectedly accurate even for values of $\delta t \sim \mathcal{O}(1)$; however, the splitting in Eq. (8) is a necessary step to perform gate decomposition of the annealing dynamics on a quantum device, as well as to perform efficient classical simulations.

## 2.3 dQA in the Tensor Network formalism

In this section, we introduce our novel tensor network framework, which allows to efficiently simulate dQA for any classical Hamiltonian in the form of Eq. (4). First, let us notice that the initial state $|\psi_0\rangle$ can be trivially represented as an MPS of bond dimension $\chi = 1$. Indeed, by setting each local tensor to $A^{[i]}(\sigma_i) = 1/\sqrt{2}$, we find

$$\langle\boldsymbol{\sigma}|\psi_0\rangle = \prod_{i=1}^{N} A^{[i]}(\sigma_i), \tag{10}$$

$|\boldsymbol{\sigma}\rangle$ being a generic state of the computational basis $\left(\hat{\sigma}_i^z|\boldsymbol{\sigma}\rangle = \sigma_i|\boldsymbol{\sigma}\rangle\right)$. Next, our goal is to rewrite the two unitaries $\hat{U}_z(\gamma_p)$ and $\hat{U}_x(\beta_p)$ as Matrix Product Operators (MPO) [20]. To begin with, $\hat{U}_x$ admits an elementary decomposition into an MPO of bond dimension $\chi = 1$:

$$\langle\boldsymbol{\sigma}'|\hat{U}_x(\beta_p)|\boldsymbol{\sigma}\rangle = \prod_{i=1}^{N} \langle\sigma_i'|e^{i\beta_p\sigma_i^x}|\sigma_i\rangle = \prod_{i=1}^{N} W^{[i]}(\sigma_i', \sigma_i),$$
$$W^{[i]}(\sigma_i', \sigma_i) = \delta_{\sigma_i', \sigma_i} \cos\beta_p + i\left(1 - \delta_{\sigma_i', \sigma_i}\right) \sin\beta_p.$$

Since here $\chi = 1$, the tensors reduce to simple scalars $W^{[i]}(\sigma_i', \sigma_i)$ (which depend on the time step angle $\beta_p$). The MPO decomposition for $\hat{U}_z$ is more challenging. First, one can factorize $\hat{U}_z$ into terms depending on a single pattern (see Eq. (4)), resulting in the following matrix element:

$$\langle\boldsymbol{\sigma}'|\hat{U}_z(\gamma_p)|\boldsymbol{\sigma}\rangle = \prod_{\mu=1}^{N_\xi} \langle\boldsymbol{\sigma}'|\hat{U}_z^\mu(\gamma_p)|\boldsymbol{\sigma}\rangle = \delta_{\boldsymbol{\sigma}', \boldsymbol{\sigma}} \prod_{\mu=1}^{N_\xi} e^{-i\gamma_p h(\xi^\mu \cdot \boldsymbol{\sigma})}. \tag{11}$$

Let us now exploit the specific form of the Hamiltonian in Eq. (4). We notice that, by definition, any such Hamiltonian depends on the spin configuration $\boldsymbol{\sigma}$ only via the following variables:

$$m^\mu(\boldsymbol{\sigma}) = \xi^\mu \cdot \boldsymbol{\sigma} = \text{ overlap between } \boldsymbol{\sigma} \text{ and } \xi^\mu,$$

or, equivalently:

$$x^\mu(\boldsymbol{\sigma}) = \frac{N - \xi^\mu \cdot \boldsymbol{\sigma}}{2} = \text{ number of bits of } \boldsymbol{\sigma} \text{ that are different from } \xi^\mu. \tag{12}$$

The latter expression is the well-known Hamming distance between $\boldsymbol{\sigma}$ and $\xi^\mu$ and, accordingly, we observe that

$$m^\mu \in \{-N, -N+2, ..., N-2, N\}, \qquad x^\mu \in \{0, 1, ..., N\}.$$

This is a key point, as it represents the only hypothesis which our construction relies on. In fact, since $x$ (for any pattern $\mu$) is an integer variable taking values in the discrete set $\{0, 1, ..., N\}$, then any function $O(x)$ can be rewritten by means of the Discrete Fourier Transform (DFT) as follows

$$O(x) = \frac{1}{\sqrt{N+1}} \sum_{k=0}^{N} \tilde{O}_k\, e^{i\frac{2\pi}{N+1}kx}, \tag{13}$$

where the Fourier coefficients are computed as

$$\tilde{O}_k = \frac{1}{\sqrt{N+1}} \sum_{x=0}^{N} e^{-i\frac{2\pi}{N+1}kx}\, O(x). \tag{14}$$

By setting $h(\xi^\mu \cdot \boldsymbol{\sigma}) = f(x^\mu(\boldsymbol{\sigma}))$, where $x^\mu(\boldsymbol{\sigma})$ is defined by Eq. 12, and using the DFT expansion reported in Eqs. (13) and (14), we can further manipulate Eq. 11 as follows:

$$
\begin{aligned}
\langle \boldsymbol{\sigma}'|\hat{U}_z^\mu(\gamma_p)|\boldsymbol{\sigma}\rangle &= \delta_{\boldsymbol{\sigma}',\boldsymbol{\sigma}} \frac{1}{\sqrt{N+1}} \sum_{k=0}^{N} \tilde{U}_{k,p}\, e^{i\frac{2\pi}{N+1}kx^\mu(\boldsymbol{\sigma})}, \\
\tilde{U}_{k,p} &= \frac{1}{\sqrt{N+1}} \sum_{x=0}^{N} e^{-i\frac{2\pi}{N+1}kx}\, e^{-i\gamma_p f(x)},
\end{aligned}
\tag{15}
$$

where the Fourier components $\tilde{U}_{k,p}$ depend implicitly on the angle $\gamma_p$, so they can be regarded as a matrix of dimension $(N+1) \times P$. Remarkably, this Fourier decomposition allows us to find an *efficient* representation of $\hat{U}_z(\gamma_p)$ as a Matrix Product Operator. This is accomplished by using Eq. 12, which can be reformulated more explicitly as

$$
x^\mu(\boldsymbol{\sigma}) = \sum_{i=1}^{N} \left( \frac{1 - \xi_i^\mu \sigma_i}{2} \right).
$$

Indeed, by identifying the wave-numbers $k = 0, 1, \dots N$ as *auxiliary* indices in the MPO formalism, we can rewrite

$$
\langle \boldsymbol{\sigma}'|\hat{U}_z^\mu(\gamma_p)|\boldsymbol{\sigma}\rangle = \prod_{i=1}^{N} W_{k_{i-1},k_i}^{[i]}(\sigma_i', \sigma_i),
\tag{16}
$$

where we defined the following tensors, diagonal by construction in the *physical* spin indices:

$$
\begin{cases}
W_{1,k}^{[i]}(\sigma_i', \sigma_i) = \delta_{\sigma_i',\sigma_i} \left( \frac{\tilde{U}_{k,p}}{\sqrt{N+1}} \right)^{\frac{1}{N}} e^{i\frac{\pi}{N+1}k(1-\xi_i^\mu \sigma_i)}, & i = 1, \\[2mm]
W_{k,k'}^{[i]}(\sigma_i', \sigma_i) = \delta_{\sigma_i',\sigma_i} \left( \frac{\tilde{U}_{k,p}}{\sqrt{N+1}} \right)^{\frac{1}{N}} e^{i\frac{\pi}{N+1}k(1-\xi_i^\mu \sigma_i)} \delta_{k,k'}, & i = 2, \cdots, N-1, \\[2mm]
W_{k,1}^{[i]}(\sigma_i', \sigma_i) = \delta_{\sigma_i',\sigma_i} \left( \frac{\tilde{U}_{k,p}}{\sqrt{N+1}} \right)^{\frac{1}{N}} e^{i\frac{\pi}{N+1}k(1-\xi_i^\mu \sigma_i)}, & i = N.
\end{cases}
\tag{17}
$$

In Eq. (16) we set $k_0 = k_N = 1$, whereas tensors are implicitly summed over repeated auxiliary indices, each of them spanning $N+1$ values. Therefore, we found that each unitary time evolution operator $\hat{U}_z^\mu(\gamma_p)$ associated to a given pattern $\mu$ can be written *efficiently* as an MPO of bond dimension $\chi = N + 1$. We remark that the four-indices tensors $W_{k,k'}^{[i]}(\sigma_i', \sigma_i)$ are diagonal also in the auxiliary indices, effectively depending only on a single auxiliary index $k$ as well as on a single physical one $\sigma_i$. However, note that they also depend on the angle $\gamma_p$ and on the pattern index $\mu$, both of which are not explicitly indicated. In light of these results, the whole dQA time evolution can be represented exactly as the 2D Tensor Network in Figure 3, corresponding to the application of a series of MPOs to the initial (trivial) MPS $|\psi_0\rangle$. This result is the starting point for our MPS-based algorithm for the classical simulation of dQA, which is summarized in the pseudo-code 1. Importantly, when contracting this 2D Tensor Network, the MPS bond dimension would increase exponentially with the number of Trotter slices P. This calls for an effective compression procedure, which is detailed in the next section: indeed, a crucial input parameter of our algorithm is the fixed maximum bond dimension of the MPS.

Let us notice that the MPO representation shown in Eq. 17 can also be exploited to write the Hamiltonian itself as an MPO of bond dimension $\chi = N + 1$. This fact is remarkable, since it allows the *exact evaluation* of the classical cost function (i.e. the expectation value of $\hat{H}_z$) over *any* MPS: in particular, when applying the Algorithm 1, we can keep track of the exact value of the cost function along the time evolution. Moreover, the representation of $\hat{H}_z$ as

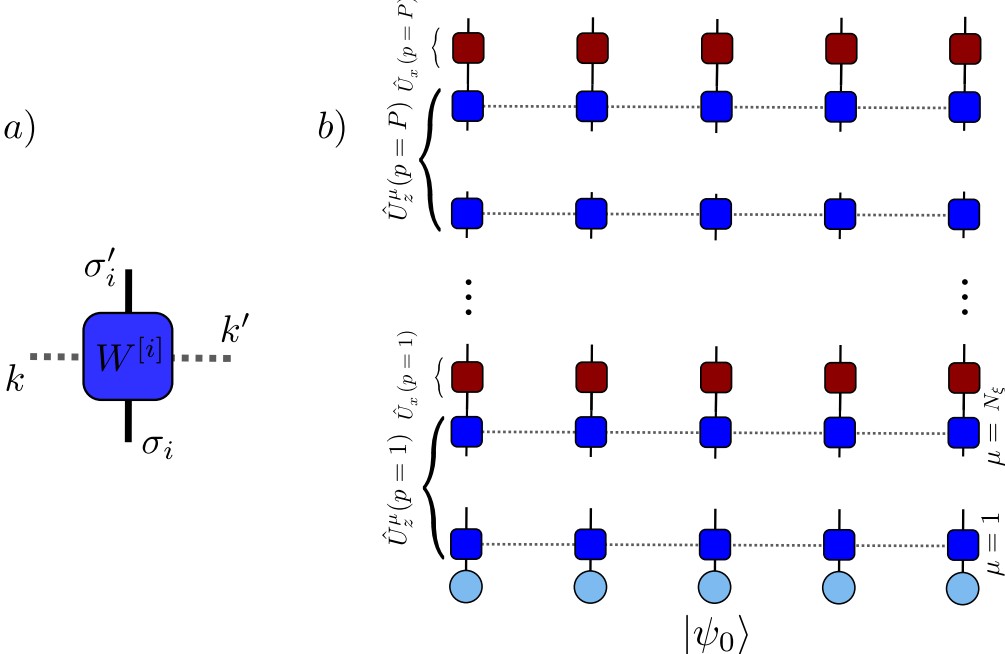

Figure 3: a) The four-indices MPO local tensor $W_{kk'}^{[i]}(\sigma_i, \sigma_i')$ of the $\hat{U}_z^\mu(\gamma_p)$ time evolution operator. Dotted grey lines represent auxiliary indices of dimension $N + 1$ ($k, k' = 0, 1, ..., N$). b) The trotterized time evolution of dQA as a 2D Tensor Network. Blue (red) squared shapes represent the MPO tensors for the decomposition of $\hat{U}_z^\mu(\gamma_p)$ $\left(\hat{U}_x(\beta_p)\right)$ time evolution operators.

MPO makes it possible to apply the DMRG algorithm, in order to find an approximate ground state of the system. However, as shown in Appendix A, this method has some issues, since the quality of the final result is strongly dependent on the initial guess provided to the DMRG optimization. Moreover, even if DMRG reaches convergence, the optimized state turns out to have a sensible overlap with only one of the degenerate ground states configurations (classical solutions). The QA protocol, instead, always reaches delocalized quantum states, having large overlaps with many different solutions (see Appendix A for details). This delocalization over clusters of classical solutions is an interesting confirmation of previous results [43, 44] in a new setting, and it may represent a winning feature for some classical optimization problems. In addition, whereas methods as DMRG are purely related to the classical simulation of many-body quantum systems, our MPS-scheme simulates dQA, providing a benchmark for Quantum Annealing experiments on real quantum devices, beyond the small system sizes reachable by means of ED methods.

Finally, it is worthwhile to mention that our tensor network representation can be easily adapted to study imaginary time evolution and the classical equilibrium properties of the corresponding classical models, i.e. without the transverse term $\hat{H}_x$.[1]

### 2.3.1 The MPS compression

A key step in our proposed algorithm is the iterative compression of the MPS wave function, reported in line 5 of the pseudo-code 1. As anticipated, this is necessary to avoid an exponential increase of the MPS bond dimension with the number of Trotter steps P. In practice, the

---

[1]Indeed, in this case, the classical equilibrium partition function will assume the form of a 2D tensor network as in Fig. 3, without the single-qubit layers.

---

**Algorithm 1** dQA with MPS

---

**Input**: the dQA parameters $\tau$, $P$, $\delta t = \tau/P$, the MPS maximum bond dimension $\chi$

1: Compute the Fourier matrix $\tilde{U}_{k,p}$ (Eq. 15) of dimension $(N+1) \times P$
2: Define the initial state $|\psi_0\rangle$ as an MPS of bond dimension $\chi = 1$ and set $|\psi\rangle = |\psi_0\rangle$
3: **for** $(p = 1, p = P, p++)$ **do**
4:     **for** $(\mu = 1, \mu = N_\xi, \mu++)$ **do**
5:         apply $\hat{U}_z^\mu(\gamma_p)$ to the current MPS $|\psi\rangle$ and compress to a lower bond dimension $\chi$
6:         update $|\psi\rangle$ setting it equal to the resulting MPS
7:     **end for**
8:     apply $\hat{U}_x(\beta_p)$ to the current MPS $|\psi\rangle$
9: **end for**

**Output**: the final optimized MPS $|\psi\rangle$

---

$a)$
$\left(\tilde{\psi}^{[i]}_{\alpha_{i-1}\alpha_i}\right)^*$

$\alpha_{i-1} \cdots \qquad \cdots \alpha_i$

$\sigma_i$

$b)$
$$\frac{\partial}{\partial(\tilde{\psi}^{[i]}_{\alpha_{i-1},\alpha_i})^*}\left(||\,|\tilde{\psi}\rangle - |\psi\rangle\,||^2\right) =$$

$$= \frac{\partial}{\partial} \left( \quad |\tilde{\psi}\rangle \quad - \quad |\psi\rangle \quad \right) = 0$$

Figure 4: The iterative compression algorithm for MPS. Dotted lines represent auxiliary indices. The compressed (uncompressed) MPS tensors are represented by violet (green) shapes. $a)$ The local compressed tensor $\tilde{\psi}^{[i]}$. $b)$ The local optimization equation 18.

compression procedure projects the dynamics on the manifold of MPS with fixed maximum bond dimension $\chi$. This projection can be achieved for instance by the iterative algorithm reported in Ref. [20,47], which is designed to solve site-by-site the optimization equation

$$\frac{\partial}{\partial(\tilde{\psi}^{[i]}_{\alpha_{i-1},\alpha_i})^*}\left(||\,|\tilde{\psi}\rangle - |\psi\rangle\,||^2\right) = \frac{\partial}{\partial(\tilde{\psi}^{[i]}_{\alpha_{i-1},\alpha_i})^*}\left(\langle\tilde{\psi}|\tilde{\psi}\rangle - \langle\tilde{\psi}|\psi\rangle\right) = 0\,, \tag{18}$$

where $|\psi\rangle$, $|\tilde{\psi}\rangle$ are respectively the uncompressed and the compressed MPS and $\alpha_{i-1}, \alpha_i$ are the local auxiliary indices. By solving these linear equations, one minimizes the Hilbert space distance between the two states with respect to the compressed local tensor $\tilde{\psi}^{[i]}_{\alpha_{i-1},\alpha_i}$, as shown in a graphical representation in Fig. 4. In practice, one iteratively performs a series of local minimizations, by sweeping along all the system sites a certain number of times $N_{sweeps}$. The computational cost of the MPS contractions involved in this procedure is $\mathcal{O}(N_{sweeps}ND^2\chi)$, $D$ and $\chi$ being the uncompressed and compressed MPS bond dimensions ($D > \chi$) [20]. In our particular case, $|\psi\rangle$ is given by the application of a unitary time evolution operator to the MPS at the previous step, i.e. $|\psi\rangle = \hat{U}_z^\mu(\gamma_p)|\phi\rangle$. In general, when an MPO is applied to an MPS, the corresponding bond dimensions are multiplied, thus we have $D = (N+1)\chi$ and the overall cost of one compression is $\mathcal{O}(N_{sweeps}N^3\chi^3)$, $\chi$ being the maximum bond dimension set as an input of the algorithm 1.

However, the compression efficiency can be greatly improved by exploiting the particular structure of the MPO in Eq. 17. To this scope, let us first observe that each operator $\hat{U}_z^\mu(\gamma_p)$

can be equivalently recast into a sum of $(N+1)$ MPOs, each of bond dimension $\chi = 1$. This fact can be seen directly from Eq. 17, by noticing that the tensors $W$ are diagonal in the auxiliary indices $k, k'$. More formally, if we define the tensors

$$W^{k;[i]}(\sigma'_i, \sigma_i) = \delta_{\sigma'_i,\sigma_i}\left(\frac{\tilde{U}_{k,p}}{\sqrt{N+1}}\right)^{1/N} e^{i\frac{\pi}{N+1}k(1-\xi_i^\mu \sigma_i)}, \quad k = 0, 1...N, \quad i = 1, 2...N,$$

we have

$$\sum_{k=0}^{N}\prod_{i=1}^{N}\left(W^{k,[i]}(\sigma'_i, \sigma_i)\right) = \delta_{\boldsymbol{\sigma}',\boldsymbol{\sigma}}\sum_k \frac{\tilde{U}_{k,p}}{\sqrt{N+1}}e^{i\frac{2\pi}{N+1}k\,x^\mu(\sigma)} = \langle\boldsymbol{\sigma}'|\hat{U}_z^\mu(\gamma)|\boldsymbol{\sigma}\rangle\,, \qquad (19)$$

namely the operator $\hat{U}_z^\mu(\gamma)$ is a sum of $N + 1$ MPOs with bond dimension $\chi = 1$. Therefore, the uncompressed MPS $|\psi\rangle = \hat{U}_z^\mu(\gamma_p)|\phi\rangle$ can be written as sum of $N + 1$ MPS $|\psi_k\rangle$, each with bond dimension $\chi$, so that Eq. 18 becomes

$$\frac{\partial}{\partial(\tilde{\psi}_{\alpha_{i-1},\alpha_i}^{[i]})^*}\left(\langle\tilde{\psi}|\tilde{\psi}\rangle - \sum_{k=0}^{N}\langle\tilde{\psi}|\psi_k\rangle\right) = 0\,. \qquad (20)$$

Now, the second term involves $N + 1$ contractions of MPS with bond dimension $\chi$. The computational cost is therefore reduced by a factor $N$, i.e. to $\mathcal{O}(N_{sweeps}N^2\chi^3)$.

### 2.3.2 Estimate of the computational cost

The computational bottleneck of algorithm 1 is the iterative compression of MPS sketched in the previous section. This is repeated $PN_\xi$ times, and each repetition is expected to have a computational cost $\mathcal{O}(N_{sweeps}N^2\chi^3)$, where $\chi$ is the maximum bond dimension. If we set $N_{sweeps} = \mathcal{O}(1)$, the overall algorithmic cost is therefore estimated to be $\mathcal{O}(PN_\xi N^2\chi^3)$ and, in the regime $N_\xi \sim N$, we expect $\mathcal{O}(PN^3\chi^3)$. A detailed evaluation of the computational cost of each step of algorithm 1 is provided in the following table. The most computationally expensive steps are 3−9, whereas the initial Fourier transforms can be performed at cost $\mathcal{O}(PN\log N)$ by means of the Fast Fourier Transform. In particular, the actual computational bottleneck is the application of each $\hat{U}_z^\mu(\gamma_p)$ to the current MPS, and the subsequent compression to the prescribed bond dimension $\chi$.

| Step | Cost |
|------|------|
| 1 | $\mathcal{O}(PN\log N)$ |
| 2 | $\mathcal{O}(N)$ |
| 3-9 | $\mathcal{O}(PN_\xi N^2\chi^3)$ |

Concerning the cost function evaluation, by exploiting the same MPO structure of Eq. 17 (with $\tilde{h}_k$ in place of $\tilde{U}_{k,p}$), once again one can rewrite $\hat{H}_z = \sum_{\mu=1}^{N_\xi}\hat{h}_\mu$ as a sum of $N_\xi(N+1)$ MPOs of bond dimension 1, instead of a single MPO of bond dimension $N_\xi(N+1)$. This allows to reduce the cost of the tensor contractions involved in the evaluation of $\langle\psi|\hat{H}_z|\psi\rangle$, that is $\mathcal{O}(N_\xi N^2\chi^3)$, instead of $\mathcal{O}(N_\xi^3 N^4\chi^3)$. If the cost function is evaluated at each step of dQA the overall cost is $\mathcal{O}(PN_\xi N^2\chi^3)$, that is of the same order of the algorithm itself.

## 2.4 MPS compilation to quantum circuits

The final output of dQA simulated via tensor networks (algorithm 1) is an optimized MPS, hopefully having large overlap with the subspace of ground states of the target Hamiltonian $\hat{H}_z$.

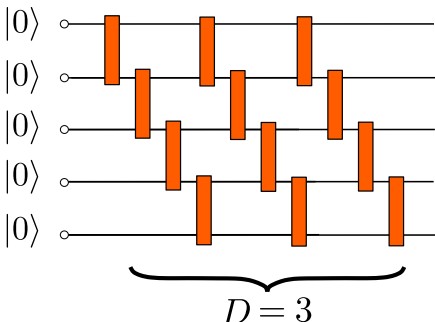

Figure 5: The general circuit structure used to "compile" the optimized MPS. $D(=3)$ is the number of staircase layers of two-qubits unitaries applied to the initial state $|\mathbf{0}\rangle$. The two-qubits unitaries (orange boxes) are optimized to yield maximum fidelity with the target MPS.

Despite our simulation being performed with purely classical resources, one can map this final result onto a quantum circuit, leading to implementations on real near-term quantum devices. This remarkable fact may set the stage to further manipulate the state with quantum resources, for example with additional hybrid quantum-classical optimizations [10]. The mapping onto a quantum circuit can be achieved by exploiting the MPS nature of the final state. It is indeed well-known that, in a $N$ qubits system, any MPS having maximum bond dimension $\chi = 2^n$ can be obtained from the trivial state[2] $|\mathbf{0}\rangle = |0...0\rangle$ by applying sequentially $N$ unitary gates, each acting (at most) on $\log_2 \chi + 1 = n + 1$ qubits [26, 48–50] (see Appendix B for additional information). These unitaries can be further decomposed into two-qubits gates. As shown in [51], the number of CNOT gates necessary for the decomposition of a single $(n + 1)$ qubits gate is $\mathcal{O}(3 \cdot 4^{n-1}) = \mathcal{O}(\chi^2)$. Thus, the optimized MPS can be recast into a quantum circuit consisting of $\mathcal{O}(N\chi^2)$ elementary two-qubits gates (such a CNOTS).

Although this proof of concept is remarkable, in order to practically "compile" the optimized MPS into a quantum circuit we shall focus on another method, reported in [48]. This employs an algorithm that iteratively optimizes two-qubits unitaries in a fixed circuit architecture, in order to maximize the fidelity with the target MPS. The geometry of the circuit is given by a fixed number $D$ of staircase layers of two-qubits gates, as represented in Fig. 5. Technicalities on the employed algorithm and results of its application are deferred to Appendix B.

## 3 Results

In this section, we analyze and discuss our numerical results on MPS simulation of dQA. Here, we focus on the benchmark p−spin model and on the binary perceptron; the same qualitative results hold for the Hopfield model, as reported in Appendix D. In the MPS simulations of the following sections, we fix a relatively small value of the bond dimension $\chi = 10$: in Appendix C, we perform an analysis of the convergence for increasing values of $\chi$. With such bond dimension, we are able to study systems up to size $N \simeq 100$ (see Sec. D), whereas ED is necessarily limited to $N \simeq 20$ due to the exponential complexity of the quantum many-body Hilbert space.

---

[2]Here, we follow the usual convention in quantum computation, identifying the spin up $|\uparrow\rangle$ (down $|\downarrow\rangle$) eigenstates of a qubit with the computational basis states $|0\rangle$ ($|1\rangle$).

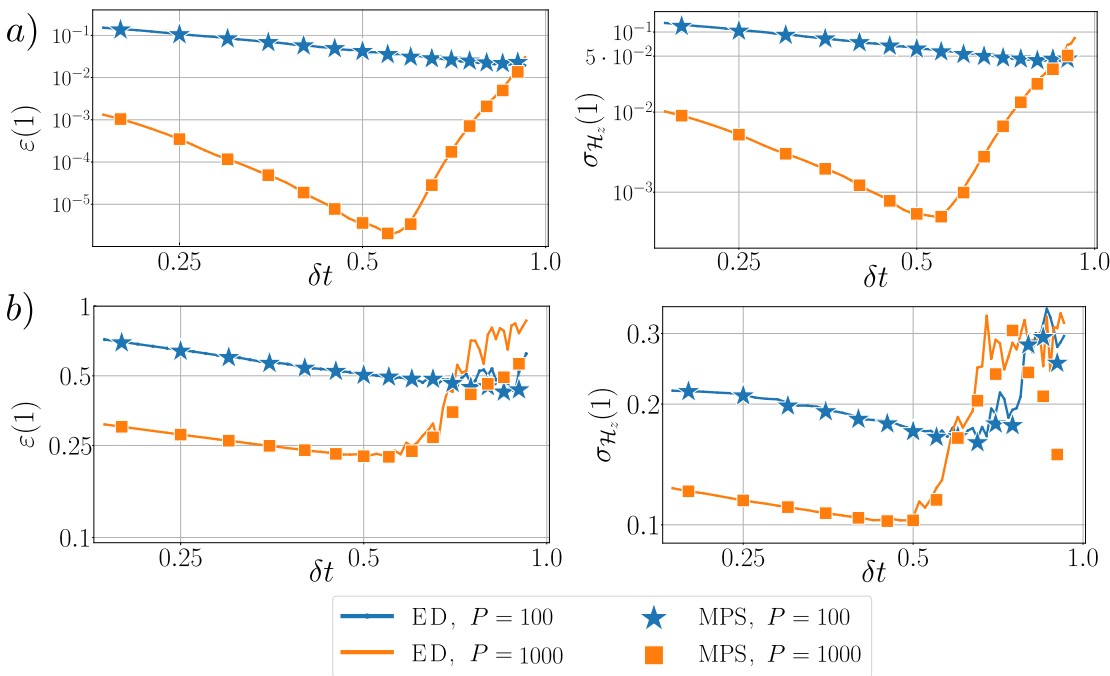

Figure 6: p–spin model. Residual energy density $\varepsilon(1)$ (left column) and residual standard deviation of the energy $\sigma_{\hat{H}_z}(1)$ (right column) as a function of the time step $\delta t$, for $a$) p = 2 and $b$) p = 3. We compare MPS ($\chi = 10$) and ED results. The system size is $N = 50$.

It is useful to introduce the energy density

$$\varepsilon(s) = \frac{\langle\psi(s)|\hat{H}_z|\psi(s)\rangle - E_{gs}}{N}, \tag{21}$$

where $|\psi(s)\rangle$ is the instantaneous wave function for a certain value of the annealing parameter $s \in [0, 1]$ and $E_{gs}$ is the ground-state energy of $\hat{H}_z$. The residual energy density at the end of the annealing schedule, defined as $\varepsilon(1)$, can be regarded as a figure of merit of dQA effectiveness. The instantaneous standard deviation of the time-dependent Hamiltonian $\hat{H}(s)$ is written as

$$\sigma_{\hat{H}}(s) = \frac{1}{N}\left( \langle\psi(s)|\hat{H}^2(s)|\psi(s)\rangle - \langle\psi(s)|\hat{H}(s)|\psi(s)\rangle^2 \right)^{1/2}, \tag{22}$$

with an analogous definition for $\sigma_{\hat{H}_z}(s)$. Notice that the residual standard deviation of $\hat{H}_z$, i.e. $\sigma_{\hat{H}_z}(1) = \sigma_{\hat{H}}(1)$, is another possible figure of merit for dQA, since it is expected to vanish if the exact ground-state of $\hat{H}_z$ is reached. Moreover, we remark that — if the dynamics is perfectly adiabatic — $\sigma_{\hat{H}}(s)$ is expected to be constantly 0 for any value of the annealing parameter $s$.

## 3.1 Benchmark (p-spin models)

As a preliminary check, we focus on the integrable p–spin model, in order to benchmark our MPS-based simulations against ED results, up to large system sizes. Let us remark that the Hamiltonian considered in Eq. 2 (with $\hat{H}_z$ given by Eq. 5) has a phase transition as a function of the annealing parameter $s$. In the thermodynamic limit $N \to \infty$, one encounters a second order phase transition for p = 2, and a first order phase transition for p > 2 [33]. In particular,

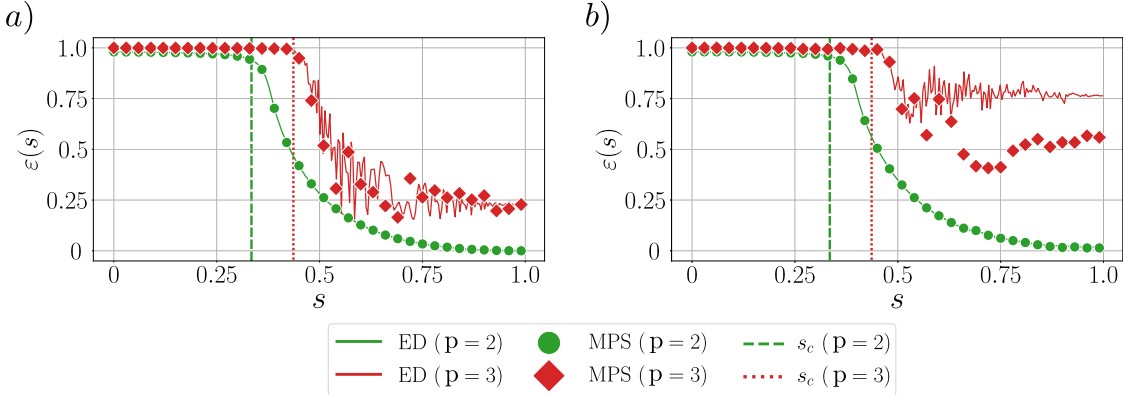

Figure 7: p−spin model. Energy density $\varepsilon(s)$ as a function of the annealing parameter $s$. Data refer to P = 1000 time steps and $a)$ $\delta t = 0.5$ (close to the minimum) and $b)$ $\delta t = 0.9$ (large Trotter errors regime). We consider both p = 2 (green) and p = 3 (red), comparing MPS ($\chi = 10$) with ED. The system size is $N = 50$.

by means of mean field calculations, one can show that the critical values are $s_c = 1/3$ for p = 2 and $s_c \simeq 0.435$ for p = 3 [33]. These phase transitions represent a challenge for Quantum Annealing, since a system is expected to stay in the instantaneous ground state of $\hat{H}(s)$ only if the total evolution time $\tau$ is (at least) inversely proportional to the square of the minimum energy gap of $\hat{H}(s)$ [31], as stated by the adiabatic theorem. Besides, it is also known that the energy gap decreases exponentially as a function of the system size $N$ at a first-order quantum phase transition, whereas the scaling is only polynomial in $N$ for a second-order transition [32]. Consequently, a QA implementation applied to finite-size systems is expected to perform worse for p = 3 than for p = 2. Let us notice that dQA can be easily simulated via ED up to large qubit numbers for the p−spin model. Indeed, the interpolating Hamiltonian $\hat{H}(s)$ in Eq. 2 commutes with the total spin operator

$$\hat{S}^2 = \frac{1}{4} \sum_{a=x,y,z} \left( \sum_{i=1}^{N} \hat{\sigma}_i^a \right)^2,$$

for any value of $s$. Since the initial state $|\psi_0\rangle$ of dQA belongs to the Hilbert space sector of maximum total magnetization, i.e.

$$\langle \psi_0 | \hat{S}^2 | \psi_0 \rangle = \langle \rightarrow \cdots \rightarrow | \hat{S}^2 | \rightarrow \cdots \rightarrow \rangle = \frac{N}{2} \left( \frac{N}{2} + 1 \right),$$

the time evolution of the system is constrained in this sector for any step $p = 1, \ldots, P$. This allows to write the time evolution operators $\hat{U}_z(\gamma_p)$ and $\hat{U}_x(\beta_p)$ as matrices of dimension $N+1 \times N+1$, and to evaluate exactly the dynamics.

In Fig. 6, we benchmark our MPS results, obtained by setting a maximum bond dimension $\chi = 10$, with ED results. This is done for p = 2, 3 models, with $N = 50$ qubits. We plot the residual energy density $\varepsilon(1)$ and the residual standard deviation $\sigma_{\hat{H}_z}(1)$ vs the time step $\delta t = \tau/P$, with a total number of steps fixed to P = 100, 1000. In all simulations we used a first-order Trotter approximation, as sketched in Section 2. As expected, ED data confirm that the final annealed state $|\psi(s = 1)\rangle$ becomes a better approximation of the ground state by increasing the time step $\delta t$ (at fixed P), until a minimum is reached, and then the protocol starts to become inaccurate due to large Trotter errors. For p = 2, the MPS results are in perfect agreement with ED for all values of $\delta t$, proving that our tensor network techniques can reproduce the exact (digitized) dynamics, despite fixing a finite bond dimension. For

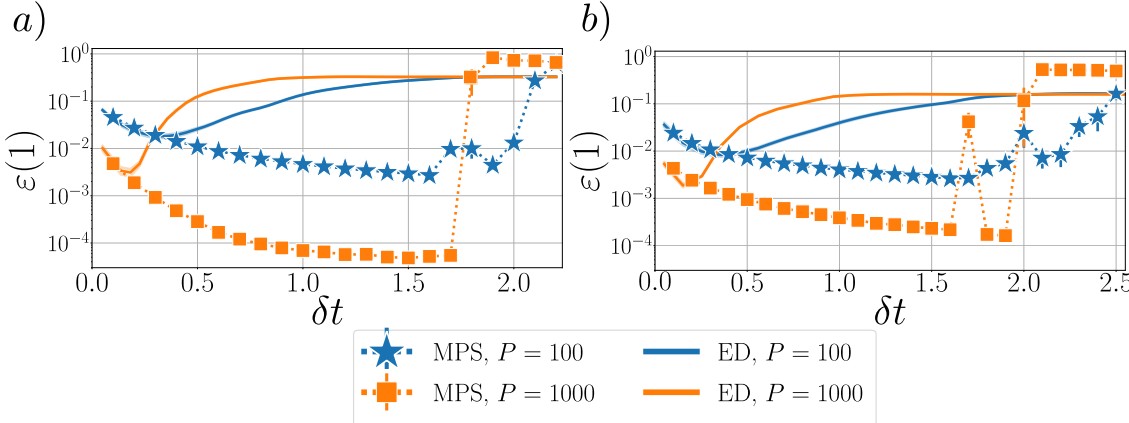

Figure 8: Binary perceptron. Residual energy density $\varepsilon(1)$ as a function of the time step length $\delta t$ for $N = 21$ and P $= 100, 1000$. Data are averaged over five different training sets, for both values of $N_\xi$ (or $\alpha = N_\xi/N$) considered: $a)$ $N_\xi = 17$ ($\alpha \simeq 0.81$) close to the SAT/UNSAT transition, and $b)$ $N_\xi = 8$ ($\alpha \simeq 0.38$) in the SAT phase. MPS results with $\chi = 10$ (full symbols) are compared with ED results (solid lines), showing good agreement for $\delta t \ll 1$, and remarkably better results for $\delta t = \mathcal{O}(1)$ (see discussion in the main text). Error bars for MPS data are given by the standard error of the means (seldom visible). Solid lines with lower opacity represent average $\pm$ the standard error of the mean for ED data.

p $= 3$, the agreement is equally good, except for the regime of large $\delta t$, which is dominated by large Trotter errors. In this case, MPS results deviate from ED, surprisingly assuming lower values of the final energy density $\varepsilon(1)$.

To summarize, in the regime of sufficiently small $\delta t$ and large values of P — i.e. where dQA approximates accurately the actual QA dynamics — our MPS techniques numerically coincide with ED simulation of dQA, thus proving an effective tool to simulate the annealing of large-size systems (at least for these benchmark models). In contrast, in the large $\delta t$ regime, dominated by large Trotter errors, the MPS dynamics can sometimes depart from ED results. However, as sketched in Fig. 2 in the Introduction, the final MPS may turn out to be a better approximation (in terms of residual energy) of the ground state, if compared to the final state obtained by exact dynamics. In this respect, let us stress that the computation of the energy density is always *exact* in our MPS framework. We will observe the same phenomenon in the next sections, for the other models considered. In Fig. 7, we also compare ED and MPS results along the annealing dynamics, for P $= 1000$. In particular, we consider two representatives values of $\delta t = 0.5$ (panel $a$) and $\delta t = 0.9$ (panel $b$), roughly corresponding to the minima of the residual energy landscape in Fig. 6 and to the regime dominated by Trotter errors, respectively. We plot the energy density $\varepsilon(s)$, as a function of $s \in [0, 1]$. For p $= 2$, we observe a perfect match of MPS and ED values of $\varepsilon(s)$, during the whole time evolution. For p $= 3$, however, the agreement is good only for $\delta t = 0.5$ (corresponding to the minimum, i.e. low Trotter errors), whereas for $\delta t = 0.9$ our MPS dynamics does not reproduce the exact digitized dynamics (at least for $s > s_c$, where $s_c$ corresponds to the phase transition point in the thermodynamic limit). Consistently with the discussion above, the MPS simulation yields substantially lower values of the energy density in the last part of the annealing (in particular for $s$ approaching to 1).

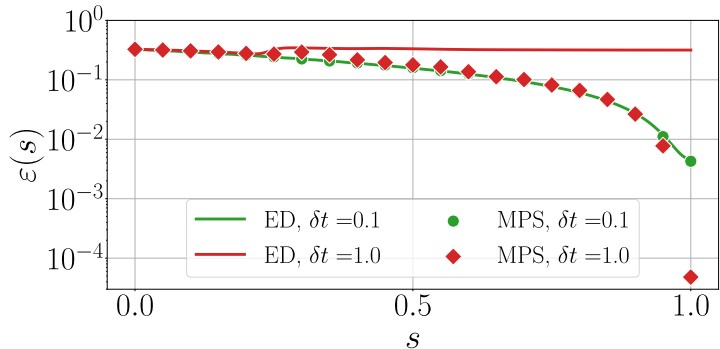

Figure 9: Binary perceptron. Energy density $\varepsilon(s)$ as a function of the annealing parameter $s$ ($N = 21$, $N_\xi = 17$, P $= 1000$). We fixed $\delta t = 0.1$ (green) or $\delta t = 1.0$ (red).

## 3.2 Binary perceptron

We now focus on the binary perceptron model, as defined by Eq. 7. First, we consider a system of size $N = 21$, so that the classical solutions/ground states can easily be found by enumeration, and MPS findings can be compared to ED results. We set $N_\xi = 17$, corresponding to $\alpha = N_\xi/N \simeq 0.81$, close to the critical value $\alpha_c \simeq 0.83$, and we run our simulations for different training sets, labeled $\{\xi^\mu\}_{\mu=1}^{N_\xi}$. In particular, we considered the same training sets used in Ref. [52], also analyzed in Ref. [44]. These were originally selected to yield highly non-convex minimization tasks, proving particularly challenging for standard Simulated Annealing techniques (i.e. Simulated Annealing suffers from an exponential slow-down, due to the trapping in meta-stable states). In the following, for conciseness, we show data averaged over five training sets in that list. However, the same results and considerations hold true for each single case in exam. In Fig. 8 (panel $a$)), we show the residual energy $\varepsilon(1)$ obtained by the MPS simulations and by ED, spanning different values of time step length $\delta t = \tau/P$. The total number of steps is fixed to P $= 100$ or P $= 1000$. Here and in the following, unless otherwise stated, MPS simulations are run by setting a maximum bond dimension $\chi = 10$. In both cases, we observe that MPS results replicate ED data for small values of $\delta t$ ($\delta t \ll 1$). On the other hand, for higher values of $\delta t$ (i.e. $\delta t \sim \mathcal{O}(1)$), ED data show an expected increase of the final energy density due to Trotter errors. On the contrary, MPS simulations surprisingly yield a further considerable decrease in the residual energy. This phenomenon supports the heuristic sketch in Fig. 2, replicating more distinctly what already observed in Fig. 6 $b$) for the p $= 3$ spin model at large values of $\delta t$. Strikingly, in this noise-dominated regime, our MPS framework still successfully performs the quantum optimization, and it yields non-trivial final quantum states that prove more efficient than those obtained by exact dQA. A detailed discussion of this phenomenon is given in Section 3.3. For completeness, we investigated whether these findings hold true also in the SAT phase, further away from the critical value $\alpha_c$. The answer is positive, as shown in Fig. 8 (panel $b$)) for $N = 21$, $N_\xi = 8$ (i.e. $\alpha \simeq 0.38$). Here, data are also averaged over five different training sets, which are randomly generated.

The remarkable difference between the two regimes of $\delta t \ll 1$ and $\delta t = \mathcal{O}(1)$ is better elucidated in Fig. 9, where we compare ED and MPS results for the two representative cases of $\delta t = 0.1$ and $\delta t = 1.0$, by plotting the energy density $\varepsilon(s)$ during the annealing. In the first case, we observe an excellent agreement between the two methods, whereas in the second case the instantaneous MPS deviates from the ED state, finally reaching considerably lower energy values. In particular, the energy density of the MPS drops by almost two orders of

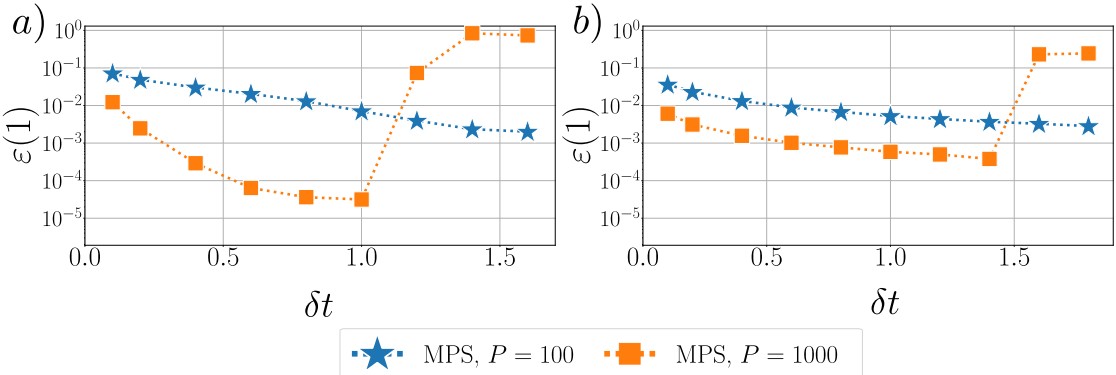

Figure 10: Binary perceptron. Residual energy density $\varepsilon(1)$ as a function of time step length $\delta t$, for a single randomly-generated training set ($N = 50$ and P $= 100, 1000$). Two values of $N_\xi$ ($\alpha$) are considered: $a$) $N_\xi = 40$ ($\alpha = 0.8$), $b$) $N_\xi = 20$ ($\alpha = 0.4$).

magnitude in the very final annealing steps (i.e. for $s \to 1$).

As stated in the Introduction, our MPS framework allows for large-scale simulations, beyond the reach of standard ED techniques. This may prove a useful benchmark for the actual implementation of quantum optimization schemes on near-term quantum devices. Here, to show the effectiveness of our methods, we address the example of a perceptron with $N = 50$ qubits, well beyond the reach of ED techniques, by studying a single randomly-generated training set. In Fig. 10 we plot the residual energy $\varepsilon(1)$ vs the time step length $\delta t$, as obtained by MPS simulations. The total number of steps is fixed again to P $= 100$ or $P = 1000$ and two similar values of $\alpha$ are considered: $\alpha = 0.8$ ($N_\xi = 40$, panel $a$) and $\alpha = 0.4$ ($N_\xi = 20$, panel $b$). Our MPS-based simulation yields a final state with low values of residual energy density, corresponding to a successful optimization. Furthermore, these results confirm the previously outlined scenario: the residual energy density keeps decreasing up to $\delta t \simeq \mathcal{O}(1)$. As a matter of fact, by increasing further the Trotter step beyond a model-dependent threshold, the algorithm enters into an unstable regime where the method fails to provide an optimal solution, and the residual energy $\varepsilon(1)$ suddenly jumps to very large values. In the discussion above, we assumed the residual energy density $\varepsilon(1)$ to be a good figure of merit of dQA effectiveness. In the following, we verify this explicitly for the case of $N = 21$ and $N_\xi = 17$, by evaluating the overlap of the final annealed MPS $|\psi(s = 1)\rangle$ with the exact ground states $\{\boldsymbol{\sigma}_a^*\}_{a=1}^{N_{sol}}$. The degeneracy of the perceptron model ranges in $N_{sol} \in [20, 160]$, depending on the particular training set in exam: here, for simplicity, we refer to the first training set with $N_{sol} = 80$. In Fig. 11 $a$) we plot (one minus) the total success probability — defined as $\sum_{\alpha=1}^{N_{sol}} p(\boldsymbol{\sigma}_a^*) = \sum_{\alpha=1}^{N_{sol}} |\langle \boldsymbol{\sigma}_a^* | \psi(s = 1) \rangle|^2$ — vs $\delta t$ for P $= 100, 1000$, confirming the same trend as previously shown in Fig. 8 (a). Additionally, in Fig. 11 $b$), we plot an histogram of probabilities $p(\boldsymbol{\sigma}_a^*)$ for a fixed value of $\delta t$ (i.e. $\delta t = 1.4$, approximately corresponding to the best MPS performance). We order the ground states $\boldsymbol{\sigma}_a^*$ such that $p(\boldsymbol{\sigma}_a^*)$ is in descending order for P $= 1000$, and we only select the 40 most probable solutions. Noticeably, the final output of dQA has a non-vanishing overlap with a number of the classical solutions of the optimization problem, i.e. the final wave function is delocalized. As already mentioned, this fact represents a remarkable difference with the DMRG algorithm, which always converges to completely localized wave functions, overlapping with a single classical solution (see Appendix A for details on DMRG results). Finally, let us mention that the final state has a non-vanishing overlap with the same set of classical solutions for both values of P, and the same is observed by comparing MPS results with ED (data not shown).

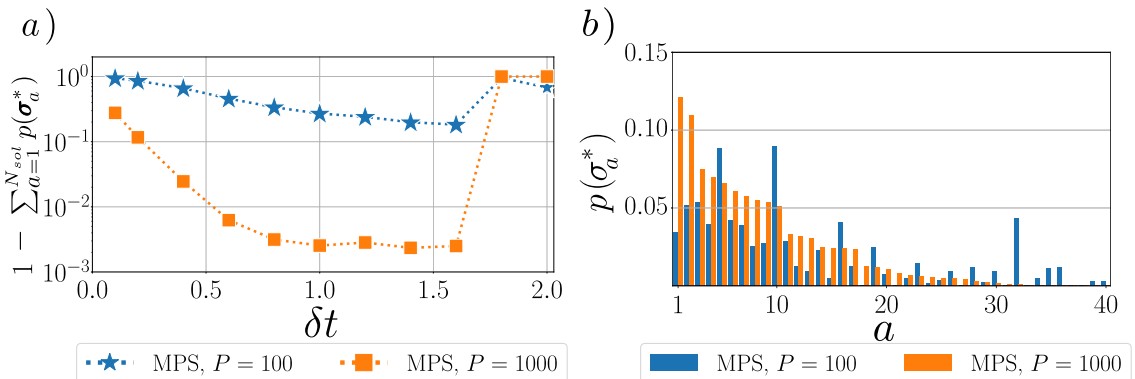

Figure 11: Binary perceptron. *a*) One minus the total success probability $\sum_{a=1}^{N_{sol}} p(\boldsymbol{\sigma}_a^*)$. The agreement with the corresponding plot for the residual energy density confirms that the latter is a good proxy for dQA effectiveness. *b*) Probabilities $p(\boldsymbol{\sigma}_a^*)$ of measuring the final state $|\psi(s=1)\rangle$ in a given solution $\boldsymbol{\sigma}_a^*$, with $\delta t = 1.4$. In both figures, data refer to a single instance with $N = 21$, $N_\xi = 17$. The same qualitative results hold for all the other training sets. Solutions are arbitrarily sorted such that $p(\boldsymbol{\sigma}_a^*)$ is in descending order for P = 1000.

## 3.3 The MPS-projection mechanism to mitigate Trotter errors

In this section, we provide some theoretical insight to explain the numerical results presented above. Let us first summarize a few preliminary concepts. The set of MPS with any fixed (finite) value of bond dimension $\chi$ constitutes a smooth manifold $\mathcal{M}_\chi$ inside the many-body Hilbert space [53], as represented in Fig. 2. The dimension of $\mathcal{M}_\chi$ is $\simeq 2\chi^2 N$, since this is the number of independent parameters in the MPS tensors, whereas the dimension of the many-body Hilbert space is exponentially greater, $2^N$. Also, $\mathcal{M}_\chi$ can be equivalently characterized as the manifold of quantum states such that the entanglement entropy between any system bipartition is *upper bounded* by $\log \chi$ [20].

Let us now go back to our MPS-based algorithm— introduced in Section 2 and summarized in the pseudo code 1 — by examining it from a geometrical point of view (as depicted in Fig. 2). We remark that the initial state of dQA, namely the fully polarized state $|\psi_0\rangle = |\rightarrow\rangle^{\otimes N}$, is an MPS with bond dimension 1 (see Eq. 10), thus it is a point belonging to the manifold $\mathcal{M}_\chi$ (for any $\chi \geq 1$). For every time step $p = 1 \cdots P$, our algorithm repeatedly projects the exact dQA dynamics on $\mathcal{M}_\chi$, after the application of each MPO $\hat{U}_z^\mu(\gamma_p)$ (corresponding to the patterns $\mu = 1, \ldots, N_\xi$). The projection is provided by the compression algorithm described in Section 2.3.1. This procedure results into an effective trajectory belonging to the manifold $\mathcal{M}_\chi$ — sketched in Fig. 2 by a gray dashed line — and it affects the entanglement content of the quantum state, constrained to be $\leq \log \chi$.

The different qualitative results obtained in the two regimes of small $\delta t \ll 1$ and larger $\delta t \sim \mathcal{O}(1)$ can be understood in terms of entanglement entropy (see Appendix C and Appendix E for details). Indeed, in the regime $\delta t \ll 1$, the entanglement of the instantaneous ED state (during the whole annealing dynamics) is relatively low, allowing for an accurate and efficient simulation of dQA with our MPS implementation, with sufficiently low values of the bond dimension $\chi$. In contrast, this is not possible for larger values of $\delta t$, since exact dQA generates large amounts of entanglement, hindering an efficient encoding with MPS techniques. Indeed, in this regime, MPS simulations largely deviate from exact dQA simulations, nevertheless surprisingly outperforming them. To clarify this peculiar aspect, we figure out that the high entanglement production and the large values of the final energy density $\varepsilon(1)$, observed in exact dQA for $\delta t \sim \mathcal{O}(1)$, are both related to Trotter errors, which can be alleviated by a

repeated projection on $\mathcal{M}_\chi$.

In previous studies (notably in Ref. [13] and [54]) it has been shown that Trotter errors largely dominate over time discretization errors in the regime of large $\delta t$. More explicitly, a comparison can be drawn between the two following protocols[3]

$$
\begin{cases}
|\phi_P\rangle = \prod_{p=1}^{P} e^{-i\hat{H}(s_p)\delta t} |\psi_0\rangle , & \text{dQA without Trotterization ,} \\[2mm]
|\psi_P\rangle = \prod_{p=1}^{P} e^{-i(1-s_p)\hat{H}_x \, \delta t} \, e^{-is_p \hat{H}_z \, \delta t} |\psi_0\rangle , & \text{dQA with Trotterization ,}
\end{cases}
\tag{23}
$$

where $p = 1, 2, \ldots, P$ and $s_p = t_p/\tau = p/P$ (as in Eq. 8). Remarkably, approximating the *exact* continuous time-ordered evolution with P discrete time steps of length $\delta t$ (dQA *without* Trotterization) turns out to be very accurate even for large time step values (i.e. $\delta t \sim \mathcal{O}(1)$), as long as P is large enough. This "robustness to time discretization" has been confirmed by several theoretical studies, such as Ref. [54], and we verify it numerically in Appendix E. On the contrary, in the same regime $\delta t \sim \mathcal{O}(1)$, dQA *with* Trotterization becomes highly inaccurate, leading to a sharp increase in the final energy density $\varepsilon(1)$. To better elucidate this fact, we can rewrite the Trotter split-up as

$$
e^{-i(1-s_p)\hat{H}_x \, \delta t} \, e^{-is_p\hat{H}_z \, \delta t} = \exp\left( -i\delta t \, \hat{H}(s_p) - \overbrace{\frac{1}{2}(\delta t)^2 s_p \left(1-s_p\right)[\hat{H}_x, \hat{H}_z]}^{\text{spurious Trotter terms}} + \cdots \right),
\tag{24}
$$

where use has been made of the lowest-order Baker-Campbell-Hausdorff expansion, neglecting $\mathcal{O}(\delta t)^3$ terms. In the regime $\delta t \sim \mathcal{O}(1)$, the large spurious Trotter terms in Eq. 24 induce non-adiabatic quantum transitions.

This scenario is confirmed by numerical simulations. In particular, we compare the two dQA protocols, *with* and *without* a first-order Trotter split-up — both of them simulated by means of ED — with our MPS scheme (we set $\chi = 10$).[4] This is done in Fig. 12 for a binary perceptron of increasing size $N$, with the parameter $\alpha$ fixed to 0.8 and $N_\xi = \alpha N$ scaling proportionally to $N$. Data are averaged over 10 random realizations of the patterns $\{\xi^\mu\}_{\mu=1}^{N_\xi}$. In panel $a$), we plot the final energy density $\varepsilon(1)$, for two representative time step values $\delta t = 0.1$ ($\delta t = 1.0$) for the small (large) time step regime, respectively. In the first case ($\delta t = 0.1$), the agreement betweeen the two dQA protocols and the MPS simulation is remarkable. Indeed, in this regime, Trotterization has negligible effects, and our MPS simulation closely resembles dQA without any Trotter split-up. In the second case ($\delta t = 1.0$), a few observations are in order. First, we notice that dQA *without* Trotterization performs even better than in the small time step regime, reaching significantly lower values of $\varepsilon(1)$. Thus, dQA would benefit, in principle, of a larger time step. However, Trotter errors here become dominant, severely spoiling the final result and confirming our analysis above. Interestingly, as sketched in Figure 2, our MPS simulation turns out to mitigate Trotter errors, providing good solutions even in this regime.

In panel $b$), we plot the von Neumann entanglement entropy for the final state.[5] In general, along the annealing protocol, this quantity is defined by

$$
S_{N/2}(s) = -\text{Tr}_{\sigma_1 \ldots \sigma_{N/2}}\big(\hat{\rho}(s)\log\hat{\rho}(s)\big), \qquad \hat{\rho}(s) = \text{Tr}_{\sigma_{N/2+1}\ldots\sigma_N}\big(|\psi(s)\rangle\langle\psi(s)|\big),
\tag{25}
$$

where $|\psi(s)\rangle$ and $\hat{\rho}(s)$ are the whole-system state and the reduced density matrix of half-system, respectively. Here, we set $s = 1$, corresponding to the final annealed states reported in

---

[3]Notice that the authors of Ref. [13] adopt a different naming convention for these protocols. Indeed, they denote dQA without Trotterization as "linear-stepQA", while dQA with Trotterization as "linear-dQA".

[4]We remark that our MPS methods allow for an efficient representation of dQA *with* Trotterization, as thoroughly explained in Section 2.

[5]We consider the so-called entanglement entropy at "half chain", i.e. between two subsystems of the same size.

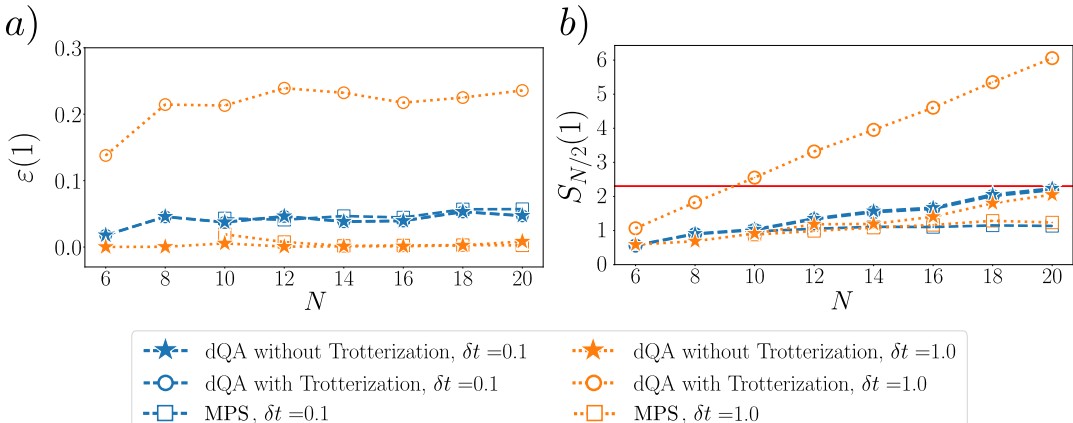

Figure 12: Binary perceptron. $a$) Final energy density $\varepsilon(1)$ and $b$) half-system entanglement entropy $S_{N/2}(1)$ of the final state, for systems of increasing size $N$. We set $\alpha = N_\xi/N = 0.8$ and P = 100. Data are averaged over 10 realizations of the random patterns, and the resulting standard deviations are smaller than the marker size. We fix two representative values of the time step, $\delta t = 0.1$ (blue) and $\delta t = 1.0$ (orange). We perform an exact simulation of dQA *with* and *without* Trotterization (see Eq. 23), and we compare these results with our MPS simulations ($\chi = 10$). The red line in panel $b$) coincides with the maximum entanglement that can be encoded into such MPS, namely $\log \chi$. In the small time step regime, Trotter errors are negligible, thus the two dQA protocols yield very similar results; moreover, they can be efficiently simulated with MPS, since the entanglement content of the final state is quite small. The qualitative picture is drastically different in the large time step regime: here, Trotter errors spoil the dQA effectiveness (large residual energy) and result in large entanglement values. Surprisingly, in this regime, MPS simulations still provide reliable low-entangled final states.

Eq. 23. For $\delta t = 0.1$, the entanglement entropy of the final state grows relatively slowly with $N$ (for both dQA protocols), allowing for efficient MPS simulations. On the other hand, for $\delta t = 1.0$, we observe that Trotter errors give rise to large amounts of entanglement entropy in the final state. Now, since our algorithm iteratively reduces the entanglement of the quantum state at each time step, we argue that it acts by iteratively projecting away the contributions given by the spurious terms introduced by the Trotter split-up, since these, as shown, are the main source of entanglement. Since the same terms are also responsible of low-quality final states (large values of final energy density), our algorithm can quite re-establish the performance of dQA *without* the Trotter split-up, thus providing a significant improvement in the final result.

In addition, a direct evidence on the MPS effectiveness in mitigating Trotter errors, and thus reproducing the actual QA dynamics, is provided in Fig. 13. Here, we fix the instantaneous state of dQA *without* Trotterization as a reference state, representing the true QA dynamics.[6] We plot the fidelity, during the annealing, between this reference state and two other states, namely the instantaneous dQA state *with* Trotter split-up — obtained either with ED, or approximated by means of our MPS framework. Remarkably, in the small time step regime (once again $\delta t = 0.1$) both fidelity values are quite close to 1, whereas in the other

---

[6]Strictly speaking, this is certainly an approximation, since this state clearly does not satisfy the ideal continuous time-ordered dynamics. However, dQA *without* Trotterization often turns out to be an outstanding approximation for a wide range of time step values (even if $\delta t \sim \mathcal{O}(1)$), as anticipated above (see Ref. [54]), and proven numerically in Appendix E.

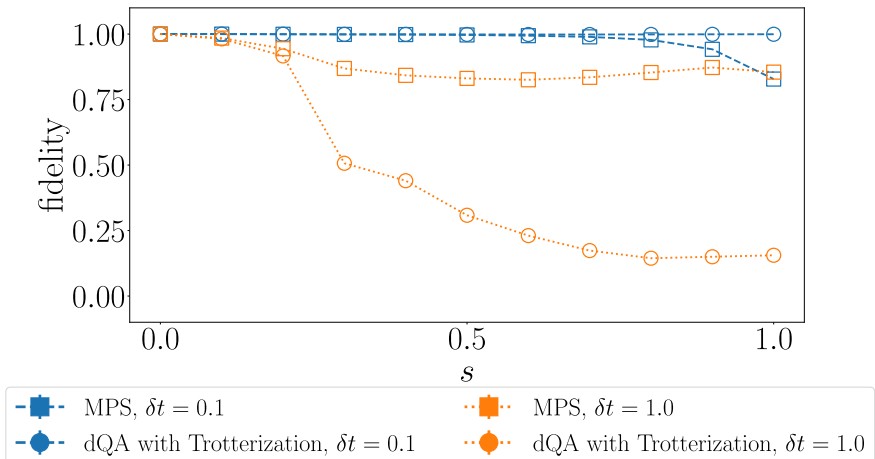

Figure 13: Binary perceptron. Fidelity between the reference state (instantaneous state of dQA *without* Trotterization) and the instantaneous state of a Trotterized dQA dynamics, either obtained with ED (circles) or with MPS simulations (squares). We fixed $P = 100$ and $\tau = 10$ (blue) or $\tau = 100$ (orange). The fidelity is plotted vs the rescaled annealing time $s = t/\tau \in [0,1]$. We set $N = 18$, $N_\xi = 14$ ($\alpha \simeq 0.78$) and we averaged over 10 realizations of the random patterns.

regime ($\delta t = 1.0$) MPS simulations prove more faithful than exact dQA with Trotterization, especially in the last stages of the annealing.

Finally, let us remark that the outlined results provide some clear insight on the correctness of our initial sketch in Fig. 2. Indeed, as shown in Fig 12 (see also Appendix C), the final state from a dQA *without* Trotterization is generally low-entangled and close to the continuous time-ordered QA results, whereas the Trotterization represents the main source of entanglement in the regime of large $\delta t$. It may therefore be convenient to approximate this Trotterized evolution as an MPS with relatively low bond dimension. As shown numerically in Fig. 13, the MPS evolved state remains closer to the annealing dynamics, if compared to the exact Trotterized evolution, confirming the qualitative representation sketched in Fig. 2.

# 4 Conclusions

We tackled the quantum optimization of a large class of Neural-Network-inspired classical minimization problems, encompassing p-spin models, and the paradigmatic binary perceptron and Hopfield models. We focused on the well-known Quantum Annealing protocol in its digitized version, and we developed a completely new, *ad hoc* approach, to simulate quantum adiabatic time evolution for these systems. This approach relies on Tensor Network methods and Matrix Product States, exploiting efficiently the available classical resources, allowing for classical simulations well-beyond the usual size limits of Exact Diagonalization.

Our Tensor Network framework could be easily employed in future work to study different non-equilibrium time evolution or the imaginary time evolution for the same class of models. Moreover, our methods could be generalized to simulate other quantum optimization algorithms, e.g. hybrid quantum-classical variational schemes such as QAOA. In this case, the Tensor Network representing the time evolution would be parameter-dependent, with the parameters being iteratively optimized by some classical routine.

We stress that the broad class of models that can be simulated in our framework includes prototypical discrete Neural Networks of significant theoretical interest in Machine Learning,

which do not admit, in general, a description in terms of few-body spin interactions. With the technological progress and the availability of real quantum devices, it is a promising route to assess the effectiveness of quantum optimization methods in this context.

Future perspectives concern the possibility to extend our methods to the optimization of more challenging discrete Neural Networks (essentially, multilayered perceptrons), as for instance Committee Machines or Treelike Committee Machines [52, 55]. This exciting chance could significantly expand our knowledge on the abilities of Quantum Computers to solve the hard optimization tasks involved in classical Deep Learning.

## Acknowledgements

We gratefully acknowledge Glen Bigan Mbeng, Nishan Ranabhat and Alessandro Santini for useful discussions. G.E.S. was partly supported by EU Horizon 2020 under ERC-ULTRADISS, Grant Agreement No. 834402, and his research has been conducted within the framework of the Trieste Institute for Theoretical Quantum Technologies (TQT).

# A    DMRG results

As outlined in Section 2, our MPS framework allows to represent any target Hamiltonian $\hat{H}_z$ in the form of Eq. 4 as an MPO of bond dimension $N + 1$ (or, equivalently, as a sum of $N + 1$ MPOs of bond dimension 1, which is computationally more efficient). In principle, this allows to apply directly the DMRG algorithm to find an MPS representation of the ground-state. To test this method, we consider again the perceptron model (Eq. 7) with $N = 21$, $N_\xi = 17$ and the same patterns $\{\xi^\mu\}_{\mu=1}^{N_\xi}$ considered in the main text. We employ the standard one-site DMRG algorithm [20], with different bond dimensions and starting from different randomly-generated MPS. For each bond dimension value $\chi = 1, 10, 20$ we run 5 different simulations. As an illustrative example, we report the DMRG data for the first training set in the following table .

| $\chi$ | Run | $\langle\hat{H}_z\rangle$ | $\chi$ | Run | $\langle\hat{H}_z\rangle$ | $\chi$ | Run | $\langle\hat{H}_z\rangle$ |
|---|---|---|---|---|---|---|---|---|
| | 1 | 0.21822 | | 1 | 0.65465 | | 1 | $-6.43 \cdot 10^{-14}$ |
| | 2 | 0.65465 | | 2 | $-3.88 \cdot 10^{-14}$ | | 3 | $-6.43 \cdot 10^{-14}$ |
| 1 | 3 | 0.21822 | 10 | 3 | $1.01 \cdot 10^{-13}$ | 20 | 2 | 0.21822 |
| | 4 | $-4.49 \cdot 10^{-14}$ | | 4 | $1.80 \cdot 10^{-13}$ | | 4 | 0.21822 |
| | 5 | $-4.48 \cdot 10^{-14}$ | | 5 | $-3.60 \cdot 10^{-14}$ | | 5 | $-6.43 \cdot 10^{-14}$ |

These results show that DMRG convergence strongly depends on the initial guess, in our case a random MPS. Indeed, since DMRG relies on a local optimization of the MPS, it can either converge to an excited state of the target Hamiltonian or, only in favourable cases, it can converge to a classical ground state. We also measure the standard deviation of the target Hamiltonian $\sigma_{\hat{H}_z}$, as defined in Eq. 22, always finding $\sigma_{\hat{H}_z} \lesssim 10^{-8}$, confirming that the final MPS is very close to be an exact eigenstate of $\hat{H}_z$. Furthermore, we measured the overlap between the MPS resulting from DMRG with the enumerated classical ground states of $\hat{H}_z$. When the minimization problem is solved properly (i.e. $\langle\hat{H}_z\rangle \approx 0$) the resulting MPS has overlap 1 (at machine precision) *only with a single solution*. This fact represents a major difference in comparison with the Quantum Annealing approach, where the final quantum state always has a finite overlap with many different classical solutions (delocalization).

# B    MPS compilation to quantum circuits

The scope of this section is to discuss the representation of an MPS as a quantum circuit, i.e. as a sequence of unitary quantum gates acting on a blank qubit register

$$|\!\uparrow\rangle = |\underbrace{\uparrow \ldots \uparrow}_{N \text{ times}}\rangle = |\underbrace{0 \ldots 0}_{N \text{ times}}\rangle = |\mathbf{0}\rangle \, .$$

In the simplest case of an MPS with bond dimension $\chi = 2$, equal to the physical dimension of the local Hilbert space, the MPS can be exactly mapped to a "staircase" of two-qubits unitary gates, acting sequentially on $|\mathbf{0}\rangle$. This mapping is well-established [26,48–50] and relies on the following fact: MPS local tensors can always be chosen as left or right isometries, which can be completed to unitary matrices, in turn decomposed into quantum gates. These tricks can be easily extended to the general case of an MPS with maximum bond dimension $\chi = 2^n$, consequently obtaining a quantum circuit defined by a sequence of unitaries acting (at most) on $\log_2 \chi + 1 = n + 1$ qubits [48–50,56]. This exact mapping of a generic MPS to a quantum circuit is schematically represented in Figure 14.

However, in order to obtain realistically feasible circuits on near-term quantum devices, one needs a decomposition into elementary gates (i.e. CNOTs and single-qubit gates) [57]. As

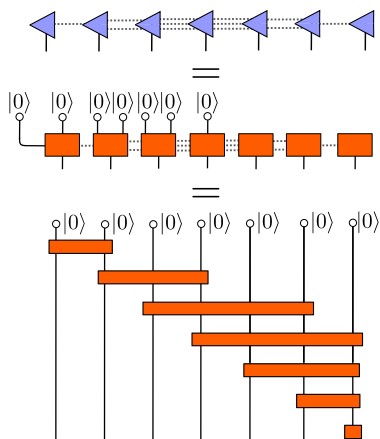

Figure 14: Exact mapping of a right-normalized MPS — with maximum bond dimension $\chi = 2^n$ ($n = 3$ in the figure) — to a staircase quantum circuit involving $N$ unitaries, each acting non-trivially on up to $\log_2 \chi + 1 = n + 1$ qubits. In the first two rows, dotted grey lines represent the MPS auxiliary bonds: a single line means that the local bond dimension $\chi_i = 2$, double lines mean $\chi_i = 2^2 = 4$, etc. These auxiliary bonds are then interpreted as multiple-qubits bonds. The original MPS isometric tensors (blue triangles) are extended to unitary matrices (orange rectangles), by completing the matrices with orthonormal rows. Therefore, they acquire extra physical indices, which are connected to $|0\rangle$ in order to recover the original tensors. In the third row, we show the final quantum circuit, yielding the original MPS as output.

anticipated in Section 2, a possible approach is to apply the previous method and then decompose each resulting unitary, involving $\log_2 \chi + 1 = n + 1$ qubits, into two-qubits unitaries. This task can be achieved with standard techniques, by using $\mathcal{O}(3 \cdot 4^{n-1})$ CNOT gates [51], and the same order of single-qubit gates. Let us notice explicitly that $3 \cdot 4^{n-1} = 0.75 \, \chi^2$, therefore the total number of employed gates scales polynomially with the MPS bond dimension. Moreover, it scales linearly with the system size $N$, since the mapping shown in Fig. 14 uses exactly $N$ unitaries.

A second possible approach relies on an approximate mapping, in which the two-qubits gates are obtained via an iterative optimization of the fidelity $\mathcal{F}(\hat{U}) = |\langle \psi | \hat{U} | \mathbf{0} \rangle|^2$, $\hat{U}$ being the unitary operator implemented by the circuit and $|\psi\rangle$ the target MPS [48]. This iterative optimization problem is schematically represented in Figure 15. In this case, we assume a fixed geometry of the circuit, composed of $D$ staircases of two-qubits unitaries (see Fig. 15). At a generic step of the iterative optimization, we fix all the two-qubits gates except for one, which is optimized. Therefore, we have to solve the maximization problem

$$\arg\max_{V} \mathcal{F}(V) = \arg\max_{V} |\langle \tilde{\psi} | V | \tilde{\phi} \rangle|^2 \, ,$$

where $V$ is the two-qubits unitary gate we need to optimize and $|\tilde{\psi}\rangle$, $|\tilde{\phi}\rangle$ are the two states obtained by applying the two remaining sets of fixed gates to $|\psi\rangle$, $|\mathbf{0}\rangle$, respectively (see Fig. 15). By defining the environment operator $E = |\tilde{\phi}\rangle \langle \tilde{\psi}|$, we have $\mathcal{F}(V) = |\text{Tr}(VE)|^2$. The inequality $|\text{Tr}(VE)| \leq ||V||_\infty ||E||_1$ applies for any square matrices $V$ and $E$, $||\cdot||_\infty$ being the operator norm and $||\cdot||_1$ the trace class norm [58]. Since we require $V$ to be unitary, $||V||_\infty = 1$, while $||E||_1 = \text{Tr}(\sqrt{E^\dagger E}) = \text{Tr}(\sqrt{\tilde{W}\Sigma W^\dagger W \Sigma \tilde{W}^\dagger}) = \text{Tr}(\Sigma)$, where $E = W\Sigma\tilde{W}^\dagger$ is the Singular Value Decomposition of the matrix $E$. Let us notice that, if we fix $V \equiv \tilde{W}W^\dagger$, then the inequality is

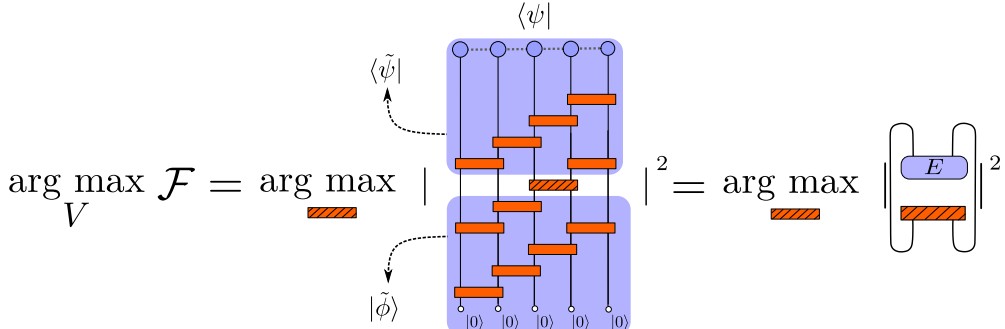

Figure 15: Approximate mapping of a given MPS to a quantum circuit made of $D$ staircases of two-qubits unitary gates (here $D = 3$). We schematically represent the optimization problem for a single two-qubits gate at a generic step of the iterative optimization algorithm. The target state $|\psi\rangle$ is represented as an MPS at the top, whereas its approximation as a quantum circuit (with fixed architecture) is obtained by applying all the required gates to the initial state $|0\ldots0\rangle$, displayed at the bottom. The two states $|\tilde{\psi}\rangle$, $|\tilde{\phi}\rangle$ are also shown explicitly.

saturated, and therefore this is an optimal choice for the local two-qubits gate. After fixing this, one can move to the next gate of the staircase, in an iterative fashion [48]. By performing an adequate number of iterations along all the gates of the circuit, this procedure is expected to reach convergence.

We perform several simulations by employing this iterative quantum circuit optimization. In our case, the target state $|\psi\rangle$ is the MPS obtained by contracting the Tensor Network structure representing the whole time evolution (see Fig. 3), by means of Algorithm 1. As an illustrative example, the following table includes results obtained for the binary perceptron ($N = 21$, $N_\xi = 17$, a single training set), by setting as target state the final state of dQA with P = 500. Different values of $\delta t$ (and hence $\tau = P\delta t$) are considered. We report the values of fidelity $\mathcal{F}$ obtained after $N_{iter} = 3000$ optimization iterations of a circuit of depth $D = 4$. We also show the corresponding value of the cost function $\langle \hat{H}_z \rangle / N = \langle \mathbf{0}|\hat{U}_{opt}^\dagger \hat{H}_z \hat{U}_{opt}|\mathbf{0}\rangle / N$ and the total success probability $\sum_{a=1}^{N_{sol}} p(\boldsymbol{\sigma}_a^*) = \sum_{a=1}^{N_{sol}} |\langle \boldsymbol{\sigma}_a^*|\hat{U}_{opt}|\mathbf{0}\rangle|^2$, where $\hat{U}_{opt}$ represents the *optimized* quantum circuit in Fig. 15.

| $\delta t$ | $\mathcal{F}$ | $\langle \hat{H}_z \rangle / N$ | $\sum_{a=1}^{N_{sol}} p(\boldsymbol{\sigma}_a^*)$ |
|---|---|---|---|
| 1.0 | 0.918 | $1.61 \cdot 10^{-3}$ | 0.941 |
| 1.2 | 0.944 | $1.63 \cdot 10^{-3}$ | 0.944 |
| 1.4 | 0.951 | $1.10 \cdot 10^{-3}$ | 0.961 |
| 1.6 | 0.894 | $1.08 \cdot 10^{-3}$ | 0.965 |

Data show that, even if the optimized Quantum Circuit does not reach particularly high fidelity $\mathcal{F}$ with the target state, anyway the final energy densities are $\mathcal{O}(10^{-3})$, and the overlap with classical solutions significantly large. This is indeed a remarkable result, for a shallow quantum circuit with $D = 4$. We mention that deeper circuits (larger values of $D$) are more challenging to optimize — possibly because the algorithm gets stuck into sub-optimal local minima — but we leave a more systematic numerical study to future work.

Let us notice that residual energy densities of $\mathcal{O}(10^{-3})$ are comparable with the best result attained by exact dQA with P = 1000 steps, and an optimal choice of $\delta t$ (see Fig. 8). Thus, we argue that by applying the Algorithm 1, and then approximating the resulting MPS $|\psi\rangle$ with a quantum circuit, one can obtain a small set of two-qubits unitary gates with a simple architecture (see Fig. 5), yielding a good quantum state, bearing large overlap with the clas-

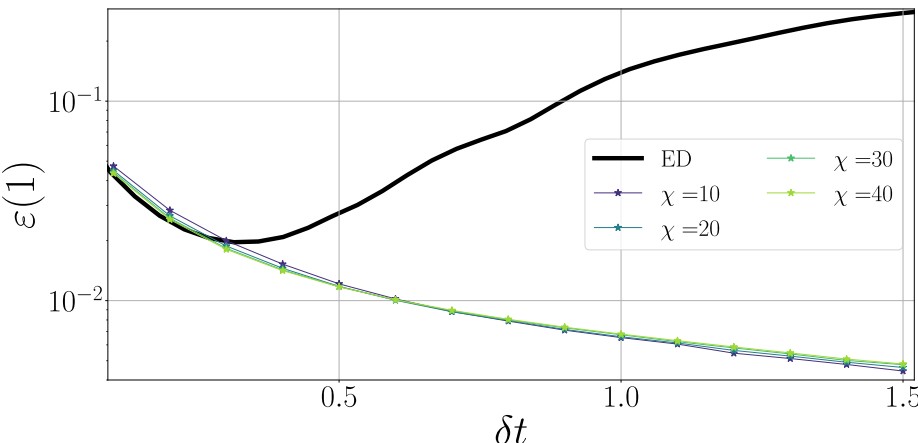

Figure 16: Binary perceptron. Residual energy density $\varepsilon(1)$ as a function of time step length $\delta t$ ($N = 21$, $N_\xi = 17$ and P = 100). MPS results for increasing values of the bond dimension $\chi$ are compared with ED results.

sical solutions. This state may represent a valuable starting point, allowing for further circuit optimization on near-term quantum devices.

## C    MPS technicalities and convergence with bond dimension

In this section, we provide extra results and details about MPS simulations. We restrict our numerics to an instance of the perceptron model (Eq. 7) for $N = 21$ spins and $N_\xi = 17$ patterns, but the same qualitative results are observed in general.

Throughout the paper, we set the bond dimension of MPS simulations to $\chi = 10$, since the validity of our methods and the main results are expected to be robust by varying the bond dimension in a reasonable range. More precisely, the bond dimension should be large enough to encode the entanglement produced by dQA in the regime of small time-step ($\delta t \ll 1$), allowing our MPS simulation to closely approximate the exact digitized dynamics. On the contrary, the large time-step regime ($\delta t = \mathcal{O}(1)$) is dominated by Trotter errors leading to high entanglement production, and we showed that MPS simulations largely deviate from ED (see Fig. 8 and Sec. 3.3 for details). Indeed, we argued that MPS simulations closely mimic a digitized dynamics *without* Trotterization, which is an excellent approximation of the continuous time-ordered dynamics of an ideal QA. Nevertheless, even in this regime, one would expect that by increasing the bond dimension, our MPS approximation of the digitized QA dynamics *with* Trotterization would converge to its exact version, implying a degradation of performance.

Here, we investigate this aspect, by performing a series of simulations for increasing values of bond dimension $\chi$, having fixed the total number of annealing steps to P = 100. Fig. 16 shows the final energy density $\varepsilon(1)$, for different values of $\delta t$, with MPS data compared with ED data (black line). In the small $\delta t$ regime, as expected, MPS data converge to ED by increasing $\chi$, confirming that our Algorithm 1 accurately simulates the exact Trotterized dQA dynamics in this regime. This is also visible in Fig. 17, where we plot the energy density difference $|\varepsilon_{MPS}(s) - \varepsilon_{ED}(s)|$ between MPS data and ED data, in the small time-step regime ($\delta t = 0.1$) for P = 100; by increasing $\chi$, this difference monotonically decreases to 0, for any value of $s$. On the contrary, in the large time-step regime $\delta t \sim \mathcal{O}(1)$ this interpretation seem-

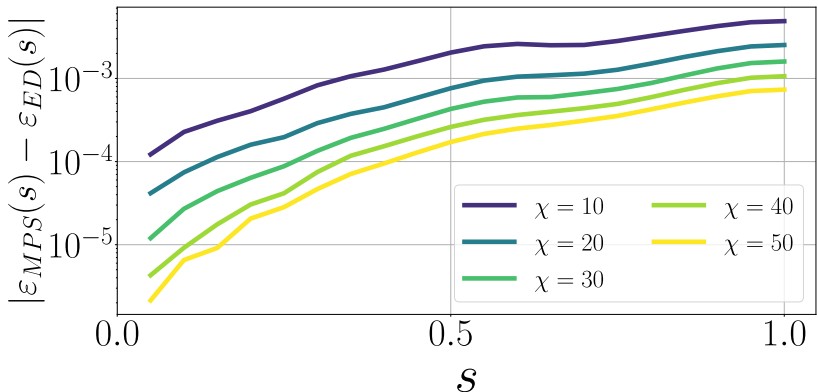

Figure 17: Binary perceptron. The energy density difference $|\varepsilon_{MPS}(s) - \varepsilon_{ED}(s)|$ between MPS and ED, as a function of the annealing parameter $s$. We set $N = 21$, $N_\xi = 17$, P $= 100$, $\delta t = 0.1$ and we explored different bond dimensions $\chi$.

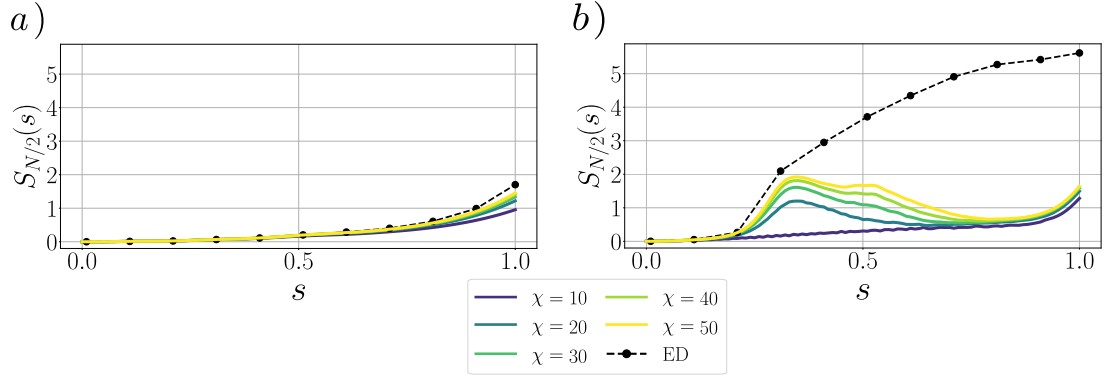

Figure 18: Binary perceptron. Half system entanglement entropy $S_{N/2}$ as a function of the annealing parameter $s$ during dQA. Here, $\delta t$ is fixed to 0.1 in panel $a$) and 1.0 in panel $b$). ED data are compared with MPS data for increasing values of the bond dimension.

ingly breaks down, as in Fig. 16 the values of $\varepsilon(1)$ are relatively stable with the increase of $\chi$, very far from ED results. We argue that the reason for this apparent inconsistency is that, in order to correctly encode the large entanglement entropy due to (unwanted) spurious Trotter terms (see Sec. 3.3), one would actually need *much larger* values of $\chi$. In the following, we show this fact more quantitatively.

In Fig. 18, we plot the entanglement entropy at half system $S_{N/2}(s)$, defined as in Eq. 25, which can be easily evaluated in the MPS framework [20]. Once again, we fix two reference values of $\delta t$ in the two regimes: $\delta t = 0.1$ (left) and $\delta t = 1.0$ (right). In the first case, MPS data show a clear convergence towards ED data upon increasing the bond dimension $\chi$. On the opposite, in the second case, MPS and ED largely deviate. In particular, the amount of entanglement produced by exact dQA reaches the value $S_{N/2}^{ED}(1) \simeq 5.62$: since the entanglement encoded by MPS is bounded by $\log \chi$, in order to encode the final steps of exact dQA in this regime, one would need (at least) $\chi \sim e^{S_{N/2}^{ED}(1)} \simeq e^{5.62} \approx 276$, far outside the range of values analyzed above.

Finally, we provide extra results on the MPS compression accuracy (see Sec. 2.3.1). Let us

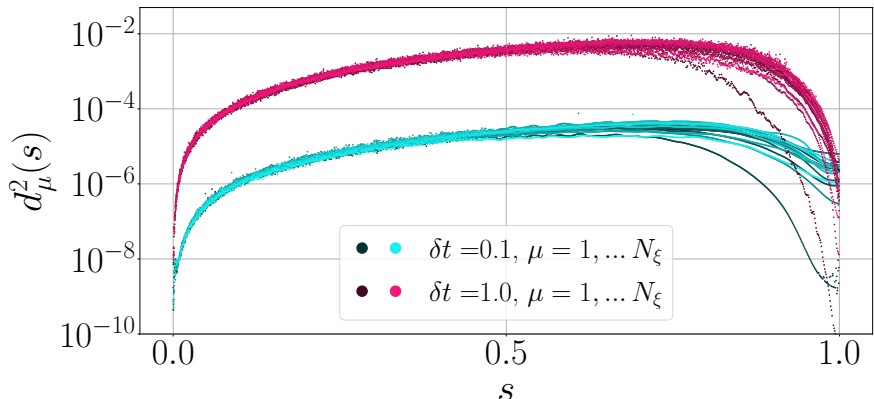

Figure 19: Binary perceptron. Hilbert space distance $d_\mu^2(s)$ between the compressed and uncompressed MPS at each step of Algorithm 1. We set $N = 21$, $N_\xi = 17$, P = 1000. Two different time steps are considered: $\delta t = 0.1$ (blue shades) and $\delta t = 1.0$ (purple shades).

address the Hilbert space distance between the compressed and uncompressed MPS at each step of the Algorithm 1, which is the same quantity being iteratively minimized in Eq. 18. This distance is written as

$$d_\mu^2(s) = || \, |\psi^\mu(s)\rangle - |\tilde{\psi}^\mu(s)\rangle \, ||^2, \qquad \mu = 1, \dots N_\xi,$$

$|\psi^\mu(s)\rangle$ being the uncompressed MPS after the application of the unitary operator $\hat{U}_z^\mu(\gamma_p)$ at a generic annealing step $s = s_p$ (with $p = 1 \cdots P$), and $|\tilde{\psi}^\mu(s)\rangle$ the MPS resulting from the compression. Thus, $d_\mu^2(s)$ is a measure of the compression accuracy. In Fig. 19, we plot $d_\mu^2(s)$ vs the annealing parameter $s \in [0,1]$, with a total number of annealing steps fixed to P = 1000. Different color shades refer to different patterns $\mu = 1 \dots N_\xi$ (the operators $\hat{U}_z^\mu(\gamma_p)$ are applied sequentially, as sketched in Fig. 3). Two values of $\delta t$ are considered: $\delta t = 0.1$ (blue shades) and $\delta t = 1.0$ (purple shades), corresponding to the two usual regimes. Notice that $d_\mu^2(s)$ takes lower values, by some order of magnitudes, for the first case (with the exception of the last part of the annealing $s \simeq 1$). The reason can be traced back to Fig. 12: in the small time-step regime, exact dQA dynamics produces low-entangled states, thus the projection into the MPS manifold is easily performed by the compression algorithm (reaching high accuracy, i.e. low values of $d_\mu^2$); for large time steps, on the contrary, the dynamics produces large amounts of entanglement, therefore the compression becomes rough (the projected state $|\tilde{\psi}^\mu(s)\rangle$ is located at larger distance from the uncompressed state $|\psi^\mu(s)\rangle$).

In the last stages of the annealing, however, the instantaneous state has projection close to one on the zero-energy ground state eigenspace, thus the unitary operators $\hat{U}_z^\mu(\gamma_p)$ act almost trivially (if the state is exactly in the ground state subspace, $\hat{U}_z^\mu(\gamma_p)$ equals the identity). This results into a sharp decrease in $d_\mu^2$ for $s$ close to 1. In the case $\delta t = 1.0$, the final values of $d_\mu^2(s)$ are smaller than for $\delta t = 0.1$, since the final state has larger overlap on the ground state eigenspace of $\hat{H}_z$, as shown in Fig. 17.

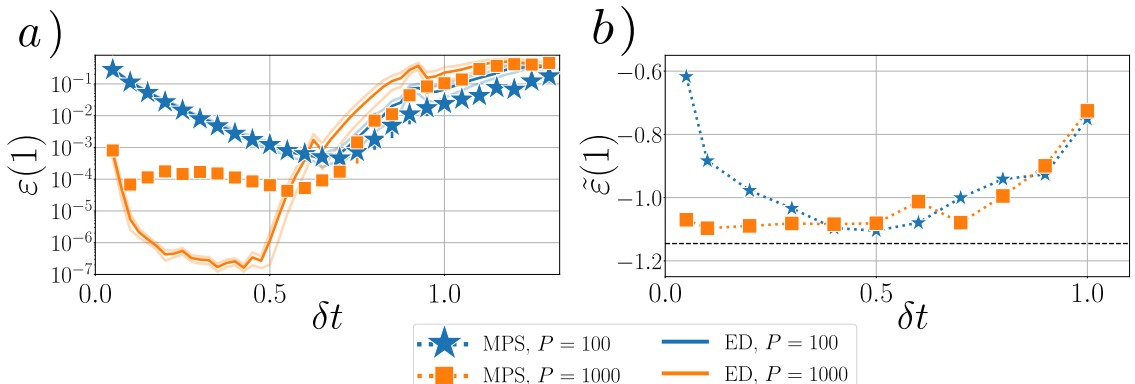

Figure 20: Hopfield model. Residual energy density as a function of time-step length $\delta t$. We set $N = 21$, $N_\xi = 2$ in panel $a$) and $N = 100$, $N_\xi = 13$ in panel $b$). In panel $a$), data are averaged over five different sets of random patterns, and MPS results (full symbols) are compared with ED results (solid lines). The (small) MPS error bars represent the standard error of the mean, often smaller than the marker size. Solid lines with lower opacity represent average $\pm$ the standard error of the mean for ED data. In panel $b$), we plot $\tilde{\varepsilon}(1)$ since the exact ground state energy is not known a priori and ED cannot be performed, due to the large system size. However, an estimate of the actual ground state energy is represented by a black horizontal dashed line (obtained by state-of-the-art classical solvers, see main text).

## D  The Hopfield model

In this section, we summarize results for the Hopfield model, defined by Eq. 6. As for the binary perceptron, we first focus on a relatively small size $N = 21$, such that exact solutions can be easily found by enumeration, and MPS results can be compared with ED. We set $N_\xi = 2$, so that $\alpha = N_\xi/N \simeq 0.095$ is close to the zero-temperature critical value $\alpha_c \simeq 0.138$, and we run our simulations for different randomly-generated training sets $\{\boldsymbol{\xi}^\mu\}_{\mu=1}^{N_\xi}$. In Fig. 20 $a$), we plot the final energy density $\varepsilon(1)$ as a function of the time step $\delta t$. As for the other models examined in the main text, we observe a close agreement between MPS and ED for small $\delta t$ (notice the log scale on the $y-$axis, therefore deviations for $P = 1000$ are actually small, of $\mathcal{O}(10^{-4})$). For $\delta t$ values of $\mathcal{O}(1)$, larger deviations are observed, with MPS generally outperforming ED. Moreover, in Fig. 20 $b$), we consider a single instance of size $N = 100$, still close to the critical point (we set $N_\xi = 13$, so $\alpha = 0.13$). In this case, the exact ground state energy is not known *a priori* and therefore we plot $\tilde{\varepsilon}(1) = \langle\psi(1)|\hat{H}_z|\psi(1)\rangle/N$. An accurate estimate of the actual minimum energy can be obtained by means of an optimized classical solver (we used the online solver http://spinglass.uni-bonn.de/): this value is represented by a black dotted line. This solver employs state-of-the-art classical optimization methods, namely a version of the branch and bound algorithm [59]. In summary, these results show that our MPS methods are effective in finding a good approximation of the target ground state even for large system sizes, far beyond the reach of ED.

## E  Time discretization versus Trotterization

The scope of this section is to provide supplementary results on the comparison between dQA *with* and *without* Trotterization (see Eq. 23). In the first place, we test numerically (for our models) a rather surprising fact, i.e. the robustness to time discretization discussed in Sec-

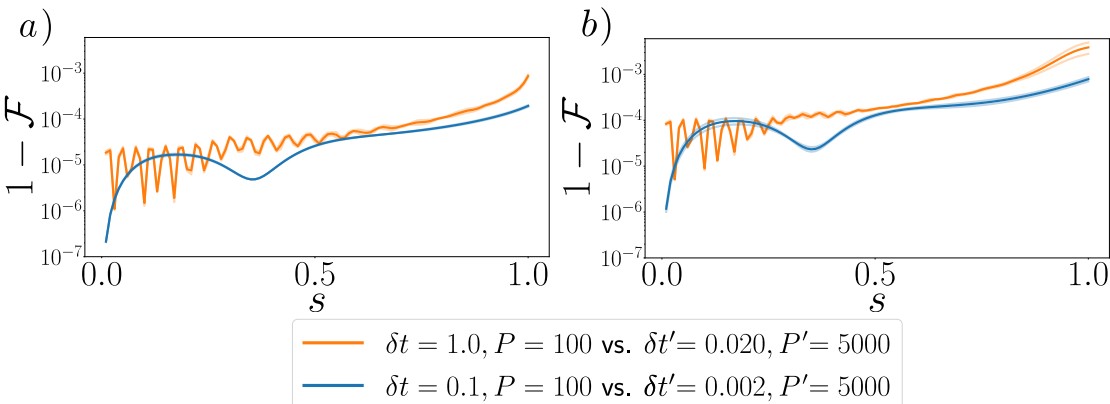

Figure 21: Binary perceptron. One minus the fidelity $\mathcal{F}(s) = |\langle\psi(s)|\psi'(s)\rangle|^2$ between the states $|\psi(s)\rangle$ and $|\psi'(s)\rangle$ both obtained with dQA *without* Trotterization, setting respectively P $= 100$, $\delta t = 0.1, 1.0$ and P$' = 5000$, $\delta t' = 0.002, 0.02$. Notice that $\delta t$P $= \delta t'$P$' = 10, 100$, meaning that the simulations are different approximations of the same continuous time evolution (with total annealing time $\tau = 10, 100$). Data are averaged over 5 realizations of the random patterns $\{\xi\}_{\mu=1}^{N_\xi}$. Lines with lower opacity represent average $\pm$ the standard error of the means. We set $N_\xi = 3$ ($\alpha \simeq 0.17$) in panel $a$) and $N_\xi = 14$ ($\alpha \simeq 0.78$) in panel $b$).

tion 3.3: the error introduced by approximating the exact continuous time-ordered evolution with P discrete time steps of length $\delta t$ is rather small, even for large $\delta t \sim \mathcal{O}(1)$. To show this, we simulate dQA *without* Trotterization (setting P $= 100$ and $\delta t = 0.1, 1.0$), and we evaluate the fidelity $\mathcal{F}$ between the evolved state $|\psi(s)\rangle$ and a reference state, ideally representing the exact time-ordered evolution. This reference state, which we dub $|\psi'(s)\rangle$, is actually obtained by running a new approximate simulation with the same values of the total annealing time ($\tau = $ P$\delta t = 10, 100$), now with a time discretization that is 50 times denser (i.e. P$' = 50$P $= 5000$, $\delta t' = \delta t/50 = 0.002, 0.02$). In practice, the simulation is performed with ED, for a perceptron model of size $N = 18$. Data are reported in Fig. 21, averaged over 5 different realizations of the random patterns. Remarkably, fidelity values are very close to 1 along the whole dynamics ($1 - \mathcal{F} < 10^{-3}$), proving that time discretization injects negligible errors in the dynamics, even for time steps as large as $\delta t = 1.0$.

An additional comparison of the two dQA methods with our MPS implementation is reported in Fig. 22. Here, for the same model and data parameters specified above, we plot the final energy density $\varepsilon(1)$ and the half-system entanglement entropy $S_{N/2}(1)$ of the final state, for different values $\delta t$. These plots are similar to those shown in Section 3, but they also include dQA *without* Trotterization. Interestingly, residual energy data show that Trotterization spoils the effectiveness of dQA at large time steps, whereas the repeated projection on the MPS manifold can (partially) restore it. The entanglement plots also confirm that Trotterization results in a strongly enhanced entanglement production, if compared to dQA *without* Trotterization; the MPS simulation significantly reduces the entanglement of the final annealed state, as already shown in Section 3.3.

Concerning the entanglement entropy of the final annealed state, let us also notice a quite peculiar aspect of our results on QA. The (average) number of solutions $N_{sol}$ is expected to decrease monotonically with the number of patterns $N_\xi$, interpolating the values $\sim 2^{N-1}$ for $N_\xi = 1$ and $\sim 0$ for $N_\xi = N\alpha_c$ (rigorously this holds for $N \to \infty$) [60]. As a consequence, one might expect the final entanglement entropy to be larger for a smaller value of $N_\xi$ (i.e. smaller $\alpha$, for fixed $N$), since the final QA wave function could acquire a non-

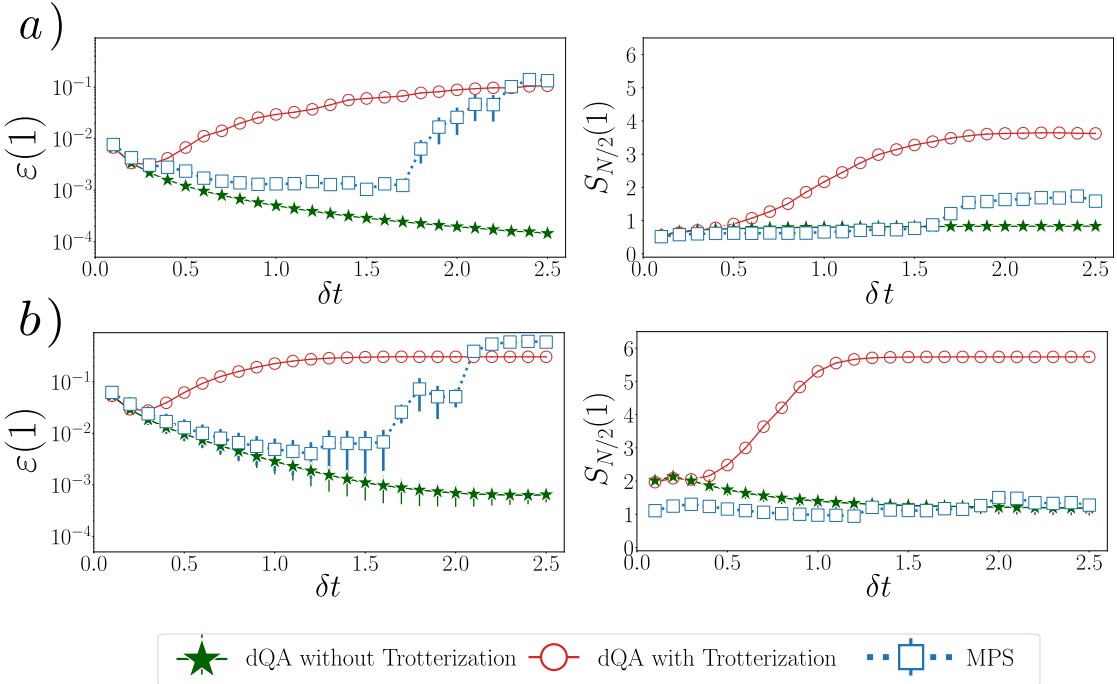

Figure 22: Binary perceptron. Final energy density $\varepsilon(1)$ (left) and half-system entanglement entropy $S_{N/2}(1)$ of the final state (right), for $N = 18$ and $N_\xi = 3\,(\alpha \simeq 0.17)$ in panel $a)$ and $N_\xi = 14\,(\alpha \simeq 0.78)$ in panel $b)$. We set P $= 100$ and we perform dQA *with* and *without* Trotterization via ED, compared with MPS results. Data are averaged over 5 realizations of the random patterns, and the resulting standard deviations are plotted as error bars (for dQA *with* Trotterization these are smaller than the marker size).

vanishing overlap with many of these solutions. However, Fig. 22 shows that, for dQA *without* Trotterization,[7] $S_{N/2}(1)$ is smaller for $\alpha = 0.17$ than for $\alpha = 0.78$ (for all values of $\delta t$).

Moreover, the half system entanglement entropy of a linear superposition of $N_{sol}$ classical states is upper bounded by $\log(N_{sol})$. For $\alpha \ll \alpha_c$, we expect $N_{sol} \sim \mathcal{O}\!\left(2^{N-N_\xi}\right)$, and thus $\log(N_{sol}) \sim (N-N_\xi)\log 2$, which finally gives $\log(N_{sol}) \sim 15 \log 2$ for panel $a)$. This is actually larger than the theoretical upper bound $N/2 \cdot \log 2 = 9 \log 2$; thus $S_{N/2}(1)$ could saturate the upper bound, but it is always observed to be much smaller. Consequently, we argue that the QA dynamics, for the problem in exam, yields a final wave function with non-vanishing overlap only with few of the many possible classical solutions. This fact is particularly relevant for our MPS approach: if QA resulted in highly entangled final states for small $\alpha$, then MPS would not be able to accurately follow its dynamics in this regime (i.e., looking back at Fig. 2, $|\psi_\tau\rangle$ would not belong to the MPS manifold $M_\chi$).

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
