# Peer review of "Quantum Annealing for Neural Network optimization problems: a new approach via Tensor Network simulations"

_SciPost Physics, doi:SciPost Phys. 14, 117 (2023)_

## Round 2 · Referee Report · Anonymous (Referee 1) · 2022-11-28

Strengths

1) The manuscript is well written, and the results are discussed in detail.

2) The proposal is excellent. I believe that the proposed method should be very important for future studies in this field, since it's so generally applicable and clearly overcomes the limitations of current exact diagonalization techniques.

3) Several optimization problems are considered. In all cases, the new method offers remarkable improvements.

4) The authors propose an efficient gate decomposition in the form of quantum circuits to implement the MPS quantum annealing strategy.

Weaknesses

1) There's little general explanation of tensor networks and matrix product states, which makes it difficult to get started reading.

Report

I think the work discussed in the manuscript is really excellent. The authors have found a very elegant way to overcome the limitations of standard matrix diagonalization techniques that are used to test quantum annealing strategies
to solve general optimization problems. They propose to use tensor networks, and in particular MPS, to encode the necessary entanglement (and nothing more), and they show that their strategy can achieve much larger system sizes than the standard methods used in the literature. The proposed strategy is tested on widely used archetypal models studied in quantum annealing (the p-spin model) and neural networks (the Hopfield model and the perceptron).
I highly recommend the publication of this manuscript in Scipost Physics once some of the concerns listed below are addressed.

Requested changes

*A brief general introduction to tensor networks and MPS techniques should be included in the manuscript. A more detailed description of the definition of the bond dimension should also be given. In addition, it should be mentioned in the main text that a study on the dependence of the results on this dimension can be found in Appendix C and the main conclusions of this study should also be discussed in the main text (in particular, how to choose chi in practice).

*I didn't understand why trotterization was used in the ED procedures if it leads to worse results for large time steps. The authors should at least explain why this approximation is considered for comparison with the performances of ED.

  • A comparison of the computation times and maximum system sizes achieved with the implementations of the different methods tested (ED with/without trotterization and MPS) should be included.

*I'm a little confused by the Hopfield example. For the capacity given, the equilibrium phase should be the metastable recovery phase. This means that the equilibrium phase of this model should be the spin glass. Why does the QA strategy recover the patterns?

  • This is a form one: The authors consider 3 different optimization problems, but only two are discussed in the main text. Throughout the text, however, there's sometimes talk of 2 problems, and sometimes of 3. In the conclusion, only the perceptron and the Hopfield are mentioned. I feel that all 3 problems could well be discussed in the main text, but if the authors prefer to leave the Hopfield problem in the appendix, the text should be revised to be consistent.

  • validity: top
  • significance: top
  • originality: top
  • clarity: top
  • formatting: excellent
  • grammar: excellent

Author:  Guglielmo Lami  on 2023-03-13  [id 3471]

(in reply to Report 1 on 2022-11-28)
Category:
answer to question

We attach a pdf containing the answer to your Report.

Attachment:

Answer_to_Report_1.pdf

---

## Round 2 · Referee Report · Marin Bukov (Referee 2) · 2022-12-13

Strengths

see report below

Weaknesses

see report below

Report

The paper "Quantum Annealing for Neural Network optimization problems: a new approach via Tensor Network simulations" by Lami et al. discusses a new framework to solve classical optimization problems in spin systems using quantum speedup provided by quantum annealing. The basic idea is to represent the state of the system and the unitary circuit as a matrix product state (MPS) of a fixed bond dimension, and work in the MPS variational manifold; the latter is necessarily area-law entangled, and hence allow to reach larger system sizes. This is a meaningful approach since both the initial x-polarized state and the target z-eigenstate (GS of a classical Ising model) are product states. The paper thus shows the existence of paths in the MPS manifold of a fixed bond dimension which connect these two states. The authors then devise a quantum circuit that allows one to implement the algorithm on quantum devices.

The results of the paper apply to a large family of classical Hamiltonians, including p-spin models, Hopfield models, and the perceptron model. The authors benchmark their results to ED on small system sizes. The computational cost of the proposed algorithm scales polynomially with the system size; this allows to perform classical simulations using MPS which go beyond ED techniques. Furthermore, the authors show that their algorithm can outperform full-space quantum annealing in the relevant regime of large Trotter step.

I think the idea behind the paper is quite novel, and the paper is very well written. Moreover the authors provide convincing analysis of the superior performance of their ansatz over a large parameter regime. Without doubt, this paper meets the criteria of SciPost physics. I recommend publication, once the authors have considered the following points.

Questions/Suggestions:

  • the authors state: "Moreover, the final annealed state |ψτ〉 — resulting from the exact QA time evolution (with τ ≫ 1) and thus expected to yield a large overlap with classical solutions — is often a low-entanglement state": is this true also when the ground state of Hz is largely degenerate? This is the case, e.g., in frustrated models; when a superposition of a large number of degenerate states is considered, the resulting state may happen to be a quantum spin liquid -- a class of topological states that possess high entanglement.

  • Fig 7b: the P=1000 data point at dt=1.7 (orange square) seems to be an outlier; did the variational algorithm get stuck in a local minimum, or what is the reason for this behavior?

  • Fig 8: what happens deeper in the UNSAT regime? Note that if exact GS cannot be reached the algorithm may still be useful in practice since in many practical cases one requires finding a single "good" solution.

  • the authors benchmark their algorithms against system sizes within the scope of ED. This is meaningful, if one wants to compare against quantum annealing. However, for N~100 there are developed tools to easily find the GS of any two-body Hz, see e.g., http://spinglass.uni-bonn.de/ . If possible, it would be nice to demonstrate one instance of a system size in the Hopfield model where the proposed algorithm outperforms maxcut (even if it doesn't find the exact GS, or it's not feasible to verify that the GS has been reached). This would require system sizes of N>120 sites or so I guess.

  • Related to the above, how much can one hope to push the system size in practice with present-day state-of-the-art classical resources? The variational optimization poses some restriction on the system size N due to the extra iteration loops.

Misc:

  • "The rest of the manuscript is organized as follows": I would make this a separate paragraph so it can be easily noticed.

  • Algorithm 1: refer to the corresponding Eqn numbers to easily locate the definition of the used quantities (e.g., $\tilde U_{k,p}$, etc.)

  • a good analysis of the computational scaling of the algorithm is provided; maybe add a separate table with the cost of the different parts of Algo 1, so the scaling can be easily located

  • Results section: it will be helpful to explicitly label each figure caption so that the model being studied is immediately visible.

Typos:

  • "suffering sensible noise": do the authors mean sensitivity to noise?
  • "for the models in exam" --> for the models we examine

Requested changes

see report above

  • validity: top
  • significance: high
  • originality: top
  • clarity: top
  • formatting: perfect
  • grammar: excellent

Author:  Guglielmo Lami  on 2023-03-13  [id 3472]

(in reply to Report 2 by Marin Bukov on 2022-12-13)

We attach a pdf containing the answer to your Report.

Attachment:

Answer_to_Report_2.pdf

---

## Round 3 · List of Changes

-We included a brief introduction to tensor networks, mainly focusing on MPS and bond dimension definitions, now provided in the Introduction.
-We clarified two points raised by the First Referee, concerning the need for trotterization in ED procedures and a comparison of computation times
and maximum system sizes achieved by means of the different numerical techniques.
-We improved the discussion on the Hopfield model, as suggested by the First Referee, in particular concerning Figure 20.
-We verified that our methods prove effective also in the UNSAT regime of the binary perceptron, as suggested by the Referee M.Bukov.
-Also stemming from M. Bukov advice, we tested our results against state-of-the-art classical optimization methods, and we better clarified which sizes we can reach with our code implementation.
-We implemented a list of minor updates, following the comments and suggestions of both Referees.

---

## Editorial Decision

published